# Learning from End User Data with Shuffled Differential Privacy over Kernel Densities

**Tal Wagner**
The Blavatnik School of Computer Science and AI
Tel-Aviv University
`tal.wagner@gmail.com`

## Abstract

We study a setting of collecting and learning from private data distributed across end users. In the *shuffled* model of differential privacy, the end users partially protect their data locally before sharing it, and their data is also anonymized during its collection to enhance privacy. This model has recently become a prominent alternative to central DP, which requires full trust in a central data curator, and local DP, where fully local data protection takes a steep toll on downstream accuracy.

Our main technical result is a shuffled DP protocol for privately estimating the kernel density function of a distributed dataset, with accuracy essentially matching central DP. We use it to privately learn a classifier from the end user data, by learning a private density function per class. Moreover, we show that the density function itself can recover the semantic content of its class, despite having been learned in the absence of any unprotected data. Our experiments show the favorable downstream performance of our approach, and highlight key downstream considerations and trade-offs in a practical ML deployment of shuffled DP.

## 1 Introduction

Collecting statistics on end user data is commonly required in data analytics and machine learning. As it could leak private user information, privacy guarantees need to be incorporated into the data collection pipeline. Differential Privacy (DP) (Dwork et al., 2006) currently serves as the gold standard for privacy in machine learning. Most of its success has been in the *central* DP model, where a centralized data curator holds the private data of all the users and is charged with protecting their privacy. However, this model does not address how to collect the data from end users in the first place. The *local* DP model (Kasiviswanathan et al., 2011), where end users protect the privacy of their data locally before sharing it, is often used for private data collection (Erlingsson et al., 2014; Ding et al., 2017; Apple, 2017). However, compared to central DP, local DP often comes at a steep price of degraded accuracy in downstream uses of the collected data.

The *shuffled* DP model (Bittau et al., 2017; Cheu et al., 2019; Erlingsson et al., 2019) has recently emerged as a prominent intermediate alternative. In this model, the users partially protect their data locally, and then entrust a centralized authority—called the "shuffler"—with the single operation of shuffling (or anonymizing) the data from all participating users. Data collection protocols in this model are designed so that the composition of shuffling over local user computations rigorously ensures DP. The appeal of shuffled DP lies in the convergence of theoretical and practical properties: mathematically, recent work has proved that shuffling can boost the accuracy of local DP to levels that may reach those of central DP (Erlingsson et al., 2019; Balle et al., 2019b; 2020b; Koskela et al., 2021; Girgis et al., 2021c; Feldman et al., 2022; 2023; Zhou & Shi, 2022). At the same time, the strictly limited functionality of the shuffler lends itself to realistic secure implementations, and a trusted shuffler can be implemented using techniques from secure computation and cryptography, like mixnets, onion routing, trusted execution environments (TEEs), and secure aggregation (SecAgg) (Ishai et al., 2006; Bittau et al., 2017; Gordon et al., 2022; Kairouz et al., 2021a;b).

There is by now a well-developed body of work on basic operations under shuffled DP, primarily summation (see Section 2.2). Work on machine learning has mostly focused on iterative settings (see Section 2.4), where distributed parties contribute local computations on their sensitive data, like

gradients, over multiple rounds of shuffled DP communication. This is compatible with *distributed (or federated) training* scenarios, in which a known set of parties collaborate in a training process that unfolds over time, typically with each contributing a local dataset and local computational resources (say, for computing local gradients). For example, the parties could be local branches of a large corporation (e.g., a bank), each holding the private data of multiple local customers.

Unfortunately, this is incompatible with *data collection* scenarios, where a "snapshot" of user data is collected in one shot from a pool of uncommitted users who hold a single training point (their own private data), which they may opt in or out of sharing, and who do not participate computationally in the training process beyond possibly contributing their data. For example, the users could be end customers of a smartphone app, prompted to privately share statistics about their app activity.

In this work, we study the *data collection* scenario under shuffled DP. We propose a private learning approach which can intuitively be seen as a shuffled DP analog of a nearest neighbor (kNN) classifier. A distributed training set of sensitive labeled data is privately collected from users, and like in kNN, subsequent test points are classified according to the most similar training examples. Since using a small number of neighbors in classification may violate their privacy, our classifier uses kernel density estimation (KDE) as a "smooth" alternative to kNN, which can be realized with shuffled DP. It thus labels test points as the class where their privately estimated density is maximized.

Moreover, our classifier produces a function representation of each class. We show this representation can be used to recover the semantics of the class—for example, a list of terms that captures the topic of a class in textual data—even though the learner did not observe any unprotected text record from the class before privacy was imposed. We refer to this as *private class decoding*.

**Our results.** Formally, we consider the following learning setting. Training data is distributed across $n$ users, each holding a single private training point $(x, c)$, where $x \in \mathbb{R}^d$ is a feature vector and $c \in [m]$ is a class label. The learner collects data from the users through shuffled DP, and uses them to construct a classifier, which can then be used to classify feature vectors $y \in \mathbb{R}^d$ from an unlabeled test set. The classifier itself needs to be private w.r.t. the collected dataset; this enables labeling an unbounded number of test points without additional loss of privacy.

To address this setting, our main theoretical result is a shuffled DP protocol for KDE estimation, that learns a private KDE function from distributed user data, which can then be used to estimate densities of test points. The utility guarantee is given in terms of the supremum mean squared error over all test points in $\mathbb{R}^d$, so that test points need not be known to the protocol in advance. The proof goes through a reduction to binary summation (abbrev. *bitsum*), which is among the most well-studied problems in shuffled DP, with a variety of available protocols to employ.

Experimentally, we evaluate our method with various combinations of kernels and bitsum protocols, yielding different trade-offs between privacy, accuracy and communication, and highlighting key downstream considerations for shuffled DP compared to central DP and local DP baselines.

## 2 BACKGROUND AND PRELIMINARIES

### 2.1 CENTRAL, LOCAL AND SHUFFLED DP

We review models of differential privacy. Let $\mathcal{X}$ be a universe of data elements. A *dataset* is an $n$-tuple $X \in \mathcal{X}^n$. Two datasets $X, X'$ are called *neighboring* if they differ on at most one coordinate. A randomized algorithm $M$, that maps an input dataset to an output from a range of outputs $\mathcal{T}$, is $(\varepsilon, \delta)$-DP if for every pair of neighboring datasets $X, X'$ and every $T \subset \mathcal{T}$, it satisfies

$$\Pr[M(X) \in T] \leq e^\varepsilon \cdot \Pr[M(X') \in T] + \delta. \tag{1}$$

In *central* DP, a single data curator holds a dataset $X \in \mathcal{X}^n$ containing the data of $n$ users, with each coordinate in the $n$-tuple $X$ representing a user. The curator runs $M$ and releases its output.

In *local* DP, each user holds her own data element, on which she runs $M$ locally, and releases its output. Here, $M$ operates on a single data element (or 1-tuple), and needs to satisfy eq. (1) for every $X, X' \in \mathcal{X}$ (every pair of single elements is neighboring). A central *analyzer* collects the already "privatized" outputs from all users and performs an aggregate computation. In this model, there is no trusted central party at all, yielding a stronger form of privacy, albeit at the cost of accuracy.

The *shuffled* DP model (Bittau et al., 2017; Cheu et al., 2019; Erlingsson et al., 2019) bridges the central and local DP models, by introducing a *limited* trusted central party—called a "shuffler"— whose only function is to anonymize (or randomly permute) the users' outputs before they are shown to the analyzer. The analyzer is considered untrusted, simialrly to local DP and unlike central DP. Formally, a shuffled DP protocol $\Pi$ consists of three randomized algorithms $\Pi = (\Pi_R, \Pi_S, \Pi_A)$:

- *Randomizer* $\Pi_R$, which maps a single element from $X$ to some sequence of messages. Each user runs $\Pi_R$ locally on her data element, and forwards the output messages to the shuffler.
- *Shuffler* $\Pi_S$, which collects the messages from all users and forwards them to the analyzer in a uniformly random order (thus, intuitively, removing sender identities).
- *Analyzer* $\Pi_A$, which receives the permuted messages from the shuffler and outputs the result of an aggregate computation.

The protocol is $(\varepsilon, \delta)$-DP in the shuffled DP model if the output of $\Pi_S$ satisfies eq. (1) (i.e., it holds with $M(X) := \Pi_S(\cup_{i=1}^n \Pi_R(X_i))$). Due to the DP post-processing property (Dwork et al., 2014), the output of $\Pi_A$ is $(\varepsilon, \delta)$-DP as well. The parties in the protocol also have access to a source of shared public randomness, which is considered publicly known, and thus cannot be exploited to compromise privacy (see Kairouz et al. (2021a)).

## 2.2 BITSUM PROTOCOLS IN THE SHUFFLED DP MODEL

Binary summation, which we refer to throughout as **bitsum**, is a fundamental and well-studied problem in DP, and particularly in shuffled DP. Each of $n$ users holds a private bit $X_i \in \{0, 1\}$, and the goal is to compute a DP estimate of the sum $S = \sum_{i=1}^n X_i$. The accuracy of a randomized estimate $\widetilde{S}$ is often quantified by its root mean squared error (RMSE), $(\mathbb{E}[(\widetilde{S} - S)^2])^{1/2}$.

A long line of work on shuffled DP bitsums (Cheu et al., 2019; Cheu & Yan, 2021; Ghazi et al., 2020b;a; 2021b; 2023) had yielded protocols whose RMSE essentially matches central DP, and is significantly better than local DP, along with additional desirable properties, like low communication and pure DP (i.e., $\delta = 0$). We will use these protocols as black-boxes and not require familiarity with their details. For completeness and intuition, we describe how they work in Appendix B.1.

## 2.3 PRIVATE KERNEL DENSITY ESTIMATION

Let $\mathbf{k} : \mathbb{R}^d \times \mathbb{R}^d \to \mathbb{R}$ be a kernel, such as the Gaussian kernel $\mathbf{k}(x, y) = \exp(-\|x - y\|_2^2)$. The kernel density estimation (KDE) map $KDE_X : \mathbb{R}^d \to \mathbb{R}$, associated with a multiset $X \subset \mathbb{R}^d$ of size $n$, is defined as $KDE_X(y) = \frac{1}{n} \sum_{x \in X} \mathbf{k}(x, y)$.

Numerous works studied KDE in the central DP model (Hall et al., 2013; Wang et al., 2016; Alda & Rubinstein, 2017; Coleman & Shrivastava, 2021; Wagner et al., 2023; Backurs et al., 2024), mostly in a setting known as *function release*. In this setting, the data curator holds all of $X$, and her goal is to release a function description $\widetilde{K}(\cdot)$ which is DP w.r.t. $X$, such that $\widetilde{K}(y)$ is an accurate estimate of $KDE_X(y)$ for every $y \in \mathbb{R}^d$. We will adapt this problem to the shuffled DP model in Definition 3.1. Our approach to this problem will use the following notion of *locality-sensitive quantization* (LSQ), recently introduced in Wagner et al. (2023) for KDE in the central DP model.

**Definition 2.1** (Wagner et al. (2023)). *Let $Q, R, S, \beta > 0$. Let $\mathcal{Q}$ be a distribution over pairs of functions $f, g : \mathbb{R}^d \to [-R, R]^Q$. We say that $\mathcal{Q}$ is a $\beta$-approximate $(Q, R, S)$-locality sensitive quantization (abbrev. LSQ) family for the kernel $\mathbf{k}$, if the following are satisfied for all $x, y \in \mathbb{R}^d$:*

- $|\mathbf{k}(x, y) - \mathbb{E}_{(f,g) \sim \mathcal{Q}}[f(x)^T g(y)]| \leq \beta$.
- $f(x)$ and $g(y)$ have each at most $S$ non-zero coordinates.

*If this holds, then $\mathbf{k}$ is $\beta$-approximate $(Q, R, S)$-LSQable. If $\beta = 0$, then $\mathbf{k}$ is $(Q, R, S)$-LSQable.*

For example, Wagner et al. (2023) observed that the Gaussian kernel is $(1, \sqrt{2}, 1)$-LSQable by random Fourier features (Rahimi & Recht, 2007), and the Laplacian and exponential kernels are $\beta$-approximate $(O(\beta^{-1}), 1, 1)$-LSQable for all $\beta > 0$ by locality sensitive hashing (Indyk & Motwani, 1998). They proved that LSQable kernels admit efficient KDE mechanisms in the central DP model. We will prove an analogous result for shuffled DP. While we draw on ideas from their central DP mechanism, our proofs will be different and self-contained.

## 2.4 Additional Related Work

Prior work on machine learning with shuffled DP has mostly focused on two iterative learning settings: distributed and federated model training (Cheu et al., 2021; Girgis et al., 2021b;a; Liu et al., 2021; Kairouz et al., 2021a), where users share privately computed gradients; and multi-armed and contextual bandits (Tenenbaum et al., 2021; Chowdhury & Zhou, 2022; Zhou & Chowdhury, 2023; Tenenbaum et al., 2023), where users share private contexts and rewards. The main difference from our setting is their iterative nature, in which the users communicate with the analyzer over multiple rounds of a shuffled DP protocol. Of these, Tenenbaum et al. (2021) is somewhat akin to us in that they also reduce their problem to bitsums, although in their case the connection is more direct as they assume binary rewards in their multi-armed bandits problem.

Beyond bitsums, there has been much work on shuffled DP protocols for integer and real summation (Cheu et al., 2019; Cheu & Zhilyaev, 2022; Balle et al., 2019b;a; 2020a; Ghazi et al., 2020b;a; 2021b; Balcer et al., 2021) and other basic operations (Balcer & Cheu, 2019; Ghazi et al., 2019; 2021a; Chen et al., 2020a; Chang et al., 2021; Scott et al., 2021; Tenenbaum et al., 2023). We discuss real summation in the context of our work in more detail in Appendix B.2.

Outside shuffled DP, a relevant work in the central DP model is Backurs et al. (2024), who also suggested a classifier that maximizes the privately computed similarity to a class. They presented results for the CIFAR-10 dataset by measuring distances to class means. Our experiments in Section 4 include the same data in a distributed setup, evaluated with our shuffled DP protocol.

## 3 Data Collection and Classification with Shuffled DP

In this section we present our private data collection and learning protocol. In Section 3.1, we discuss certain practical considerations with shuffled DP, that would inform the design of our method. In Section 3.2, we give our shuffled DP result for KDE, which is the main building block in our classifier. In Section 3.3, we use it to privately learn a classifier from collected user data.

### 3.1 Practical Considerations with Shuffled DP

**User counts.** A key limitation of shuffled DP is that protocols are required to know in advance the number of participating users $n$. Technically, the noise added by each local randomizer $\Pi_R$ generally decreases with $n$. This is crucial for boosted accuracy, albeit if some users drop out, the protocol fails to meet its DP guarantee for the remaining users. This limitation may be acceptable in the *distributed training* scenario from Section 1, where a predetermined group of parties is expected to collaborate on training and reliably execute the protocol. However, in our *data collection* scenario, it would make less sense to assume that the number of participating users is known in advance. We will therefore designate a preliminary communication round for allowing users to opt into participation.

**Privacy threat models.** There are several possible places to impose DP in an ML pipeline. Ponomareva et al. (2023) outline three options, from the most stringent to most lenient form of privacy: *input/data-level DP*, where the adversary has access to the data used to train the ML model; *model-level DP*, where the adversary has full access to the weights of the trained model; and *prediction-level DP*, where the adversary has access only to model outputs when presented with test points.

We will consider the first two of these options, adapted to shuffled DP. Input/data-level DP means the adversary sees all communication sent to the analyzer (equivalently, the analyzer itself is the adversary). Note that communication from the users to the shuffler is never exposed (that would void the premise of shuffled DP); in practice, this line of communication is implemented cryptographically, exploiting on the restricted nature of the shuffler (Kairouz et al., 2021a). However, the adversary can see all communication between the shuffler and the analyzer, as well as all direct communication (if any) between the users and the analyzer. We refer to this as the *communication-threat model*. In model-level DP, a weaker adversary sees only the trained model released by the analyzer after the protocol execution is complete; we refer to this as the *model-threat model*. These threat models are typically not differentiated in prior work on shuffled DP, since they often coincide; however, in our case, the trained model would leak less privacy than the communication used to learn it.

---

**Algorithm 1:** Shuffled DP KDE protocol from bitsums

---

**Global initialization** // *all data here is public*

    **input:** shuffled DP bitsum protocol $\Pi$;
             $(Q, R, S)$-LSQ family $\mathcal{Q}$; integer
             $I > 0$

    **for** $i = 1, \ldots, I$ **do**

        $(f_i, g_i) \leftarrow$ independent sample from $\mathcal{Q}$
        // *using shared/public randomness*

        **for** $j = 1, \ldots, Q$ **do**

            $\Pi_{ij} \leftarrow$ independent instance of $\Pi$

    **publish:** $(f_i, g_i)$ for all $i = 1, \ldots, I$

**Randomizer** // *each user runs this locally with private randomness*

    **input:** private data point $x \in \mathbb{R}^d$

    **for** $(i, j) \in [I] \times [Q]$ **do**

        $b_{ij} \leftarrow \text{Bernoulli}(0.5(1 + (f_i(x))_j/R))$

        $\Gamma_{ij} \leftarrow$ run the randomizer of $\Pi_{ij}$ on $b_{ij}$

        **for** *message* $\gamma$ *in* $\Gamma_{ij}$ **do**

            send $(\gamma, (i, j))$ to the shuffler

**Analyzer** // *runs after the shuffler*

    **input:** shuffled sequence of messages $\widetilde{\Gamma}$
            from $n$ users

    **for** $(i, j) \in [I] \times [Q]$ **do**

        $\widetilde{\Gamma}_{ij} \leftarrow$ empty sequence

    **for** *message* $(\gamma, (i, j))$ *in* $\widetilde{\Gamma}$ **do**

        append $\gamma$ to $\widetilde{\Gamma}_{ij}$

    **for** $(i, j) \in [I] \times [Q]$ **do**

        $\widetilde{B}_{ij} \leftarrow$ run the analyzer of $\Pi_{ij}$ on $\widetilde{\Gamma}_{ij}$

        $\widetilde{F}_{ij} \leftarrow (2\widetilde{B}_{ij} - n)R$

    **publish:** $\widetilde{F}_{ij}$ for all $i, j$

**KDE Query** // *runs on the analyzer's published output arbitrarily many times*

    **input:** query point $y \in \mathbb{R}^d$

    **return:** $\frac{1}{nI} \sum_{i=1}^{I} \sum_{j=1}^{Q} \widetilde{F}_{ij} \cdot (g_i(y))_j$

---

**Bit-width and discretization.** Kairouz et al. (2021a) emphasize that in practice, the shuffler implementation often requires modular arithmetics for cryptographic secure aggregation. Therefore, the shuffled DP protocol's numerical values must be discretized, and its bit precision (called *bit-width*) needs to be explicitly bounded. Neglecting to account for the bit-width may lead to impractical communication costs and to larger errors (due to discretization) than a real-valued analysis predicts. We will therefore incorporate discretization into our protocol and account for it in the error analysis.

## 3.2 SHUFFLED DP KDE FROM BITSUM PROTOCOLS

We now present the theoretical backbone of our private learning approach, a shuffled DP protocol for KDE. We start by defining the KDE problem in the shuffled DP model.

**Definition 3.1** (shuffled DP KDE). *In the shuffled DP KDE problem, a dataset $X \in (\mathbb{R}^d)^n$ of $n$ points in $\mathbb{R}^d$ is distributed across $n$ users, one point per user. The goal is to devise a shuffled DP protocol $\Pi_{\text{KDE}}$ in which the analyzer releases a function description $\widetilde{K}(\cdot)$, required to be $(\varepsilon, \delta)$-DP w.r.t. $X$. The supremum root mean square error (abbrev. supRMSE) of the protocol is defined as*

$$\text{supRMSE}(\Pi_{\text{KDE}}) := \sup_{y \in \mathbb{R}^d} \sqrt{\mathbb{E}\left[\left(\widetilde{K}(y) - KDE_X(y)\right)^2\right]}.$$

Our main technical result is the following theorem, which is a reduction from KDE to bitsum protocols in the shuffled DP model, for kernels with the LSQ property defined in Definition 2.1. The resulting shuffled DP KDE protocol is given in Algorithm 1.

**Theorem 3.2.** *Let $\mathbf{k}$ be a $\beta$-approximate $(Q, R, S)$-LSQable kernel (cf. Definition 2.1). Suppose we have an unbiased $(\varepsilon_0, \delta_0)$-DP bitsum protocol $\Pi$ in the shuffled DP model, with RMSE $\mathcal{E}_\Pi$. Then, for every $\delta' > 0$ and integer $I > 0$, Algorithm 1 is a shuffled DP KDE protocol, which is $(\varepsilon, \delta)$-DP in the communication-threat model, where $\varepsilon = \varepsilon_0 S(e^{\varepsilon_0 S} - 1)I + \varepsilon_0 S\sqrt{2I \ln(1/\delta')}$ and $\delta = IS\delta_0 + \delta'$, with supRMSE $\sqrt{4\beta^2 + I^{-1} \cdot 16R^4 S(S + (\mathcal{E}_\Pi/n)^2)}$. The protocol has optimal bit-width 1.*

Note that $\varepsilon, \delta$ take the familiar "advanced composition" form (Dwork et al., 2014) of $I$ instances of an $(\varepsilon_0 S, \delta_0 S)$-DP mechanism. The proof of Theorem 3.2 goes by showing that the LSQ coordinates can be discretized essentially without loss of accuracy, and invoking the bitsum protocol on the discretized coordinates, using a careful probabilistic analysis to bound the overall supRMSE from their individual RMSEs. It is given in full in Appendix A.1. Appendix A.1.7 also discusses additional variants of Theorem 3.2, for bitsum protocols whose error guarantee is given in other terms than the RMSE (like Cheu et al. (2019)), or that achieve pure DP (like Ghazi et al. (2020a; 2023)).

As a concrete instantiation of Theorem 3.2, we get the following result for the Gaussian kernel, by plugging the shuffled DP protocol from Ghazi et al. (2020b) as the bitsum protocol, and random Fourier features (Rahimi & Recht, 2007) as the LSQ family. The proof is in Appendix A.2. For completeness, the protocol for this special case is fully detailed in Algorithm 2 in the appendix.

**Theorem 3.3** (shuffled DP Gaussian KDE). *There are constants $C, C' > 0$ such that the following holds. Let $\delta \in (0, 1)$ and $\varepsilon \leq C \log(1/\delta)$. For every $\alpha \geq C' \sqrt{\log(1/\delta)}/(\varepsilon n)$, there is an $(\varepsilon, \delta)$-DP Gaussian KDE protocol in the shuffled DP model (under the communication-threat model) with $n$ users and inputs from $\mathbb{R}^d$, which has: supRMSE $\alpha$, user running time $\min(O(d/\alpha^2), \tilde{O}(d + 1/\alpha^4))$, expected communication of $\tilde{O}(1/\alpha^2)$ bits per user, expected analyzer running time $O(n/\alpha^2)$, KDE query time $\min(O(d/\alpha^2), \tilde{O}(d + 1/\alpha^4))$, and optimal bit-width $1$.*

### 3.3 PRIVATE LEARNING, CLASSIFICATION AND CLASS DECODING

We now describe our private learning approach for classification and class decoding. Recall that each user holds a private data point $x \in \mathbb{R}^d$ and a corresponding label $c \in [m]$.

**Learning.** The learner will aim to learn a KDE function representation per class, using the shuffled DP protocol from Theorem 3.2. As discussed in Section 3.1, this requires knowing in advance the number of participating users per class. We therefore start with a preliminary communication round designated to privately obtain these counts. This could be done with an off-the-shelf shuffled DP histogram protocol (e.g., Ghazi et al. (2020b)); however, this again requires prior knowledge of the total number of users. To avoid this chicken-and-egg issue, we will use vanilla local DP for the preliminary communication round. It is a stronger form of privacy that requires no prior knowledge, and its accuracy, while degraded, is still sufficient for the simple task of private user counts.

Formally, let $\varepsilon_0, \delta_0, \varepsilon_{\mathrm{lbl}} > 0$ be privacy parameters. Learning proceeds as follows. First, each user locally protects her label $c$ and publishes a privatized label $\tilde{c}$, using $m$-ary randomized response (Kairouz et al., 2014; 2016). Thus, $\tilde{c}$ is set to $c$ with probability $e^{\varepsilon_{\mathrm{lbl}}}/(e^{\varepsilon_{\mathrm{lbl}}} - 1 + m)$, and to a uniformly random label from $[m] \setminus \{c\}$ otherwise. This ensures that $\tilde{c}$ is $\varepsilon_{\mathrm{lbl}}$-DP, without shuffling.

Based on the published labels, the learner groups the users into their reported classes, and publishes the count of users $\tilde{n}_{\tilde{c}}$ in each reported class $\tilde{c} \in [m]$. These counts are $\varepsilon_{\mathrm{lbl}}$-DP by post-processing, and thus safe to publish. The users in each reported class then execute the shuffled DP KDE protocol in Algorithm 1, using $\varepsilon_0, \delta_0$ as the privacy parameters for each instance of the bitsum protocol $\Pi$. The learner acts as the analyzer in all these protocols, and through them learns an approximate KDE function $\widetilde{K}_c(\cdot)$ for each label $c \in [m]$. From Theorem 3.2 and from basic DP composition, we have the following privacy guarantee:

**Corollary 3.4.** *The above learning protocol is $(\varepsilon, \delta)$-DP in the model-threat model, and $(\varepsilon + \varepsilon_{\mathrm{lbl}}, \delta)$-DP in the communication-threat model, where $\varepsilon, \delta$ are given by $\varepsilon_0, \delta_0$ as stated in Theorem 3.2.*

**Classification.** The learner classifies a test point $y \in \mathbb{R}^d$ as the class where its private density estimate is maximized, namely as $c_y = \mathrm{argmax}_{c \in [m]} \widetilde{K}_c(y)$. We refer to this as the *highest density class* (HDC) classifier. It can be viewed as generalizing the $k$-nearest neighbor (kNN) classifier, where the density of $y$ w.r.t. class $c$ is measured by the number of its $k$-nearest neighbors labeled $c$, and of the nearest class center (NCC) classifier (Papyan et al., 2020), where the density is measured by the distance between $y$ to the mean of all data points labeled $c$. Note that the kNN classifier is incompatible with DP, since its output relies on a small number of training points, while the NCC classifier is a special case of HDC, which we include in the experiments in the next section.

**Private class decoding.** To illustrate class decoding, Suppose that the data points are embeddings of text documents (even though the notion extends to other data modalities as well). Let $V \subset \mathbb{R}^d$ be a fixed public "vocabulary", say the embeddings of all words in an English dictionary. To "decode" a class $c$, the learner returns the top few vocabulary words $v \in V$ that maximize $\widetilde{K}_c(v)$. The goal is for those words to capture and convey the semantic meanings of texts from class $c$.

To underline the distinction between classification and class decoding: classification relies on the ability of the collection of functions $\{\widetilde{K}_c\}_{c \in [m]}$ to produce meaningfully rankable scores over the different classes for a *fixed input* $y \in \mathbb{R}^d$; class decoding relies on the ability of each *fixed function* $\widetilde{K}_c$ to produce meaningfully rankable scores over a large collection of inputs $V \subset \mathbb{R}^d$. In general, we

expect a learned representation of a class to encompass its semantic meaning and to be decodable. The particular challenge in shuffled DP is that the representation was learned without observing any raw training example from the class, i.e., prior to imposing differential privacy on the training data.

## 4 EXPERIMENTS

We evaluate our method with several combinations of kernels and bitsum protocols. Our code is enclosed in the supplementary material and available online.

**Datasets.** We use three textual datasets and one image dataset:

- *DBPedia-14* (Zhang et al., 2015): Text documents containing summaries of Wikipedia articles. Training examples: 560K, test examples: 70K, classes: 14, task: topic classification.
- *AG news* (Zhang et al., 2015): Text documents containing news articles. Training examples: 120K, test examples: 7.6K, classes: 4, task: topic classification.
- *SST2* (Socher et al., 2013): Sentences extracted from movie reviews. Training examples: 67.3K, test examples: 1.82K, classes: 2, task: sentiment classification (positive/negative).
- *CIFAR-10* (Krizhevsky, 2009): Images from different object categories. Training examples: 50K, test examples: 10K, classes: 10, task: depicted object classification.

The datasets are embedded in $\mathbb{R}^d$ using standard pretrained models. The textual datasets are embedded into 768 dimensions with the SentenceBERT "all-mpnet-base-v2" model (Reimers & Gurevych, 2019). CIFAR-10 is embedded into 6144 dimensions with the SimCLR "r152_3x_sk1" model (Chen et al., 2020b), pre-trained on Imagenet (these are the same embeddings used in Backurs et al. (2024) for their central DP experiment). All embedding vectors are normalized to unit length. We note that the datasets are not included in the pretraining set of the respective embedding models used to embed them, ensuring that the pretraining set does not leak privacy in our experiments.

**Kernels.** We experiment with two kernels that fit into the framework of Theorem 3.2:

- **Gaussian:** $\mathbf{k}(x, y) = \exp(-\|x - y\|_2^2)$. As noted in Section 2.3, it is $(1, \sqrt{2}, 1)$-LSQable by letting the functions in $\mathcal{Q}$ be random Fourier features, leading to Theorem 3.3.
- **IP:** The inner product kernel $\mathbf{k}(x, y) = x^T y$. Since our embeddings are normalized, it is trivially $(d, 1, d)$-LSQable. It is also $(1, \sqrt{d}, 1)$-LSQable by standard dimensionality reduction arguments (see Appendix A.3), which leads to better parameters in Theorem 3.2. Note that for a subset $X'$ of training points, the IP KDE at $y$ is $\frac{1}{|X'|} \sum_{x \in X'} y^T x = y^T (\frac{1}{|X'|} \sum_{x \in X'} x)$. Thus, the HDC classifier labels $y$ by the most similar class mean, as the NCC classifier discussed in Section 3.3.

To equalize the computational costs of the two kernels, we set the number of repetitions $I$ in Algorithm 1 to $d$ (the embedding dimension). Since the embeddings have unit length, there is no need to clip the vectors at a hyperparameter (as in Kairouz et al. (2021a); Backurs et al. (2024)) for the IP kernel, nor to optimize a bandwidth for the Gaussian kernel (it is just set to 1).

**Bitsum protocols.** We use three shuffled DP bitsum protocols from the literature, each optimal in a different measure — efficiency, accuracy and privacy, respectively (see also Appendix B.1):

- **RR**: The classical randomized response protocol, as adapted to shuffled DP by Cheu et al. (2019). This protocol has optimal communication efficiency of a single one-bit message per user.[1]
- **3NB**: The correlated noise protocol of Ghazi et al. (2020b), which has asymptotically optimal accuracy. We call it 3NB since it relies on three samples from a negative binomial distribution.[2]
- **Pure**: The pure DP ($\delta = 0$) protocol of Ghazi et al. (2023). The other protocols use $\delta > 0$.[3]

**Privacy parameters.** We follow the guidelines given in Ponomareva et al. (2023), who cite current machine learning deployments of DP as using $\varepsilon$ generally between 5 to 15, and advocate for $\varepsilon \leq 10$ as an acceptable privacy regime. We use $\varepsilon \in (0, 10)$ to protect the training point $x \in \mathbb{R}^d$ with $(\varepsilon, \delta)$-shuffle DP, and $\varepsilon_{\text{lbl}} \in \{3, 5, 7, 10\}$ to protect the label $c \in [m]$ with $(\varepsilon_{\text{lbl}}, 0)$-local DP. We use $\delta = 10^{-6}$ for DBPedia-14 and AG news, and $\delta = 10^{-5}$ for SST2 and CIFAR-10, accounting for the

---

[1]See Section 4.2 for a detailed discussion on communication costs.

[2]See $\psi_1, \psi_2, \psi_3$ in Algorithm 2 in the appendix.

[3]To maintain purity in Algorithm 1 when composing instances of Pure, they are composed with "basic" (pure) rather than "advanced" (approximate) DP composition (Dwork et al., 2014). See Appendix A.1.7.

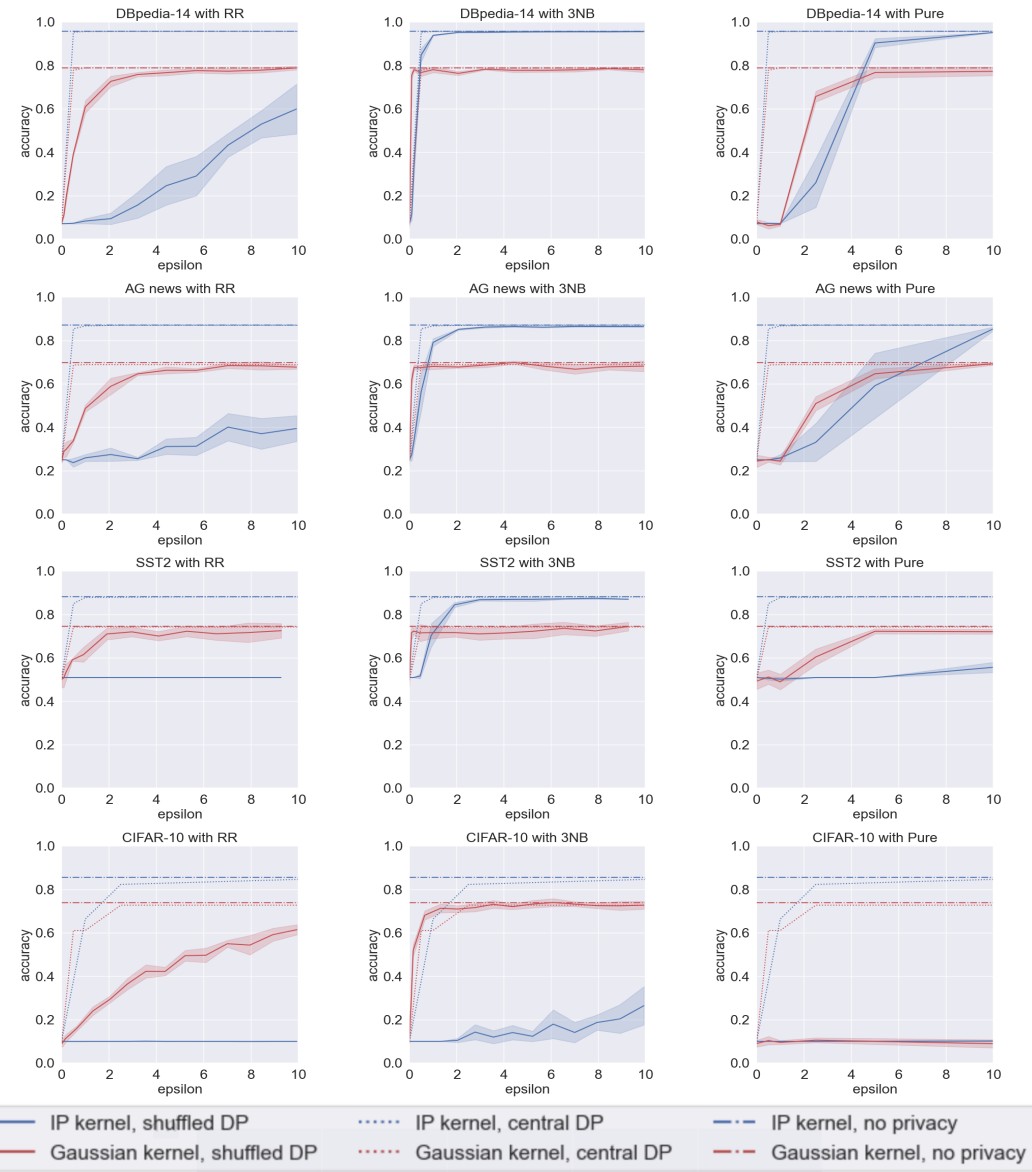

Figure 1: Classification results with $\varepsilon_{\text{lbl}} = 5$

different dataset sizes. For RR and 3NB, the $\delta$ "budget" in Theorem 3.2 is split equally between the advanced composition parameter $\delta'$ and the total $IQ\delta_0$ term of the bitsum protocol instances.

## 4.1 PRIVATE CLASSIFICATION RESULTS

We evaluate the HDC classification accuracy for each combination of kernel and bitsum protocol, {Gaussian, IP} × {RR, 3NB, Pure}, on each the dataset. Figure 1 shows results for $\varepsilon_{\text{lbl}} = 5$ (solid lines). Results for other values of $\varepsilon_{\text{lbl}}$ are similar and appear in the appendix (Figures 5 to 8).

As points of reference, we include the following two baselines for each kernel:

- $(\varepsilon, \delta)$-*central DP (dotted lines)*: HDC with the bitsum protocols in Algorithm 1 replaced by the standard Gaussian DP mechanism (see Section A in Dwork et al. (2014)).
- *No privacy (dash-dot lines)*: HDC with the bitsum protocols replaced by exact summation.

In the appendix (Figure 4), we also include an ablation against a local DP baseline.

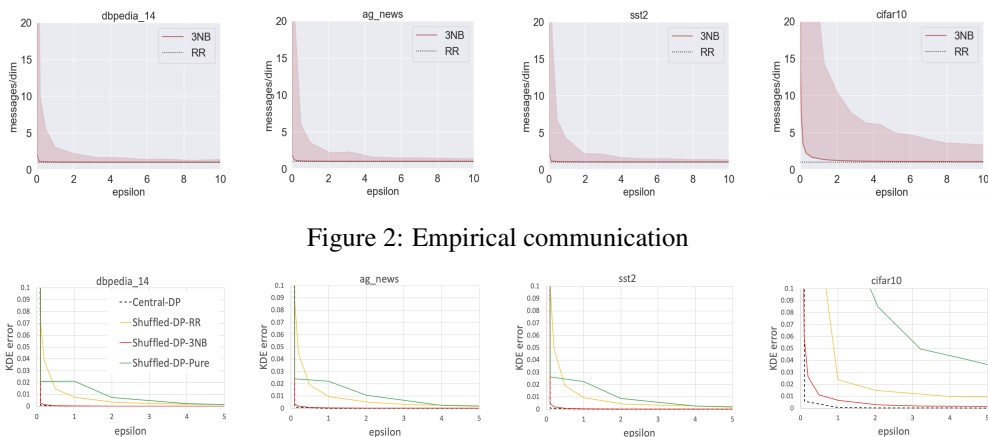

Figure 2: Empirical communication

Figure 3: Gaussian KDE accuracy

The results exhibit consistent behavior. Without privacy, the IP kernel outperforms the Gaussian kernel in all settings. This corroborates the known effectiveness of the NCC classifier on neural embeddings (see Papyan et al. (2020)). Moreover, central DP closely matches the corresponding non-DP downstream accuracy already at small values of $\varepsilon$, which corroborates a similar empirical finding reported in Backurs et al. (2024).

Under shuffled DP, however, this behavior varies in different settings. The Gaussian kernel is more resilient to errors than IP, and thus matches its central DP and non-DP HDC performance at broader parameter regimes. As a result, it often achieves better overall accuracy than IP, even though its baseline (non-DP) accuracy is lower. This phenomenon is more expressed the more error-prone the setting is – both with lower privacy budgets (lower $\varepsilon$), and when the bitsum protocol is less optimized for accuracy (i.e., RR and Pure vs. 3NB). The upshot is that the shuffled DP model, due its more delicate interplay between communication, privacy and accuracy compared to the central DP and non-DP settings, may require different and more error-resilient mechanisms for better downstream performance, particularly under tighter privacy and communication constraints.

## 4.2 COMMUNICATION COST

The communication cost of our learning protocol depends on that of the bitsum protocol used within it. Specifically, for both the Gaussian and IP kernels, the communication cost is as follows:

- With RR: each user sends exactly $d$ messages.
- With 3NB: the number of messages sent by each user is a random variable (different per user), with expectation $(1 + o(1))d$. (The $o(1)$ term vanishes as either $n$ or $\varepsilon$ grows.)
- With Pure: the number of messages sent by each user is a random variable (different per user), with expectation $O(d^2 \log(n)/\varepsilon_0)$.

With all three protocols, each message is of size $\lceil \log_2(d) \rceil + 1$ bits.

Figure 2 displays the empirical number of messages on each dataset, for RR (whose communication is constant, as per above) and 3NB (whose communication is a random variable). Note while the cost of 3NB is asymptotically near-similar to RR, in practice its cost can be a few times larger, which may be significant in applications. Pure sends orders of magnitude more messages (as per above), which may render it impractical in tight communication settings, and cannot fit on the same plots.

## 4.3 PRIVATE KDE RESULTS

We also directly evaluate Theorem 3.3 for the standalone task of private Gaussian KDE (without subsequent classification). The results are shown in Figure 3, with accuracy measured over 1K random queries from the query set of each dataset. They show that the KDE error generally tracks with the downstream classification accuracy reported above, with 3NB being the most accurate variant with error vanishing nearly as fast as central DP, followed by RR and Pure.

Table 1: Private class decoding results with $\varepsilon \approx 3.2$ and $\varepsilon_{\mathrm{lbl}} = 5$

| Dataset | Class | Bitsum | Gaussian KDE class decoding | IP KDE class decoding |
|---|---|---|---|---|
| DBPedia-14 | Company | RR | vendors, gencorp, servicers | firesign, wnews, usos |
| | | 3NB | molycorp, newscorp, mediacorp | companys, alicorp, interactivecorp |
| | | Pure | ameritech, alicorp, newscorp | alibabacom, oscorp, companies |
| | Film | RR | biopic, movie, screenplay | kaptai, kakhi, kaloi |
| | | 3NB | filmography, vanya, ghostbusters | movie, filmography, screenplay |
| | | Pure | filme, movie, videodrome | movie, film, filmmakers |
| AG news | Sports | RR | vizner, runnerups, dietrichson | ongeri, grandi, zarate |
| | | 3NB | injury, semifinalists, finalists | semifinalists, championship, standings |
| | | Pure | pensford, rematches, undefeated | chauci, teammates, nith |
| | Business | RR | repurchases, downtrend, equitywatch | sneed, timesnews, anxiousness |
| | | 3NB | enrononline, investcorp, comcorp | stockholders, nasdaq, marketwatchcom |
| | | Pure | corporations, consolidations, consolidated | merger, divestiture, stockholders |
| SST2 | Negative | RR | beguile, inception, shallow | manipulating, uncouple, dissects |
| | | 3NB | melodrama, rawness, blandness | comedy, tastelessness, uneasiness |
| | | Pure | chumminess, meaningfulness, mootness | absurdities, chastisement, absurdity |
| | Positive | RR | kindliness, pleasantness, entertaining | enjoyments, academie, amusements |
| | | 3NB | salacious, movie, majestic | salaciousness, theatricality, memorability |
| | | Pure | spiritedness, spirited, perspicacious | exorcisms, fairytales, revisiting |

## 4.4 PRIVATE CLASS DECODING RESULTS

We perform class decoding, as described in Section 3.3, on the three textual datasets. As the public vocabulary $V$ we use GloVe 6B (Pennington et al., 2014), consisting of 400K words extracted from public sources. Rather than using the embeddings from Pennington et al. (2014), we embed the terms in $\mathbb{R}^d$ with the same pre-trained SentenceBERT model used to embed the datasets. Then, for each class $c \in [m]$ of each dataset, we rank all vocabulary terms according to their density as reported by the function $\widetilde{K}_c(\cdot)$ privately learned by our shuffled KDE protocol for that class, and report the top-3 scoring terms. We repeat this for every combination of kernel and bitsum protocol.

We make no attempt at quantifying a measure of class decoding performance, since semantic relatedness is inherently somewhat subjective, and may furthermore depend on external knowledge (for example, when the "artist" or "athlete" classes are decoded into names of specific artists or athletes). Rather, our goal in this experiment to is gain qualitative insight into what the shuffled DP KDE protocol succeeds in learning, despite its lack of access to unprotected training examples, and to complement the quantitative classification accuracy results.

Table 1 includes decoding results with $\varepsilon \approx 3.2$ and $\varepsilon_{\mathrm{lbl}} = 5$ (the $\varepsilon$ values are slightly different across the datasets and bitsum protocols, due to the use of different $\delta$ values and composition theorems, as detailed earlier in this section). It only includes some classes from each dataset, due to space limits; results for all classes, and for other values of $\varepsilon$, are in the appendix.

Qualitatively, in settings where classification accuracy is nontrivial (per Figure 1), the class decoding results in Table 1 also yield vocabulary words that are aligned with the topic of the class. This demonstrates that the class representations learned by shuffled DP KDE protocol capture the semantic meaning of the classes, and preserve the ability to rank not only inter-class similarities (as needed for classification), but also intra-class similarities (as needed for decoding a specific class).

## 5 CONCLUSION

We showed how to use shuffled DP for "one-shot" data collection and learning from an undetermined pool of uncommitted end users, in contrast to prior work on ML with shuffled DP, which mostly focused on collaborative distributed training across committed parties over time. Due to the desirable accuracy of shuffled DP, our method is able to learn intricate data semantics while adhering to a distributed notion of privacy. Our experimental results highlight practical downstream considerations related to the delicate interplay between privacy, accuracy and communication cost.

Future work would explore further ways to deploy shuffled DP in ML pipelines, and extend to more challenging settings, such as privately and continuously monitoring end user data over time.

ACKNOWLEDGMENTS

We thank the anonymous reviewers for helpful feedback. Work supported by Len Blavatnik and the Blavatnik Family foundation and by an Alon Scholarship of the Israeli Council for Higher Education. Author is also with Amazon. This work is not associated with Amazon.

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

# A PROOFS

## A.1 PROOF OF THEOREM 3.2

We restate the theorem for convenience:

**Theorem A.1** (Theorem 3.2, restated). *Let* **k** *be a $\beta$-approximate $(Q, R, S)$-LSQable kernel (cf. Definition 2.1). Suppose we have an unbiased $(\varepsilon_0, \delta_0)$-DP bitsum protocol $\Pi$ in the shuffled DP model, with RMSE $\mathcal{E}_\Pi$. Then, for every $\delta' > 0$ and integer $I > 0$, Algorithm 1 is a shuffled DP KDE protocol, which is $(\varepsilon, \delta)$-DP in the communication-threat model, where $\varepsilon = \varepsilon_0 S(e^{\varepsilon_0 S} - 1)I + \varepsilon_0 S \sqrt{2I \ln(1/\delta')}$ and $\delta = IS\delta_0 + \delta'$, with supRMSE $\sqrt{4\beta^2 + I^{-1} \cdot 16R^4 S\left(S + (\mathcal{E}_\Pi/n)^2\right)}$. The protocol has optimal bit-width 1.*

The proof proceeds in three steps: (i) discretize the LSQ coordinates of $f_i(x)$ locally at each user from $[-R, R]$ to $\{-R, R\}$, using randomized rounding to maintain the LSQ property; (ii) use the bitsum protocol to estimate the sum of each discretized coordinate (with shifting and scaling to turn bitsums into $\pm R$-sums); (iii) use the LSQ property together with the RMSE bound of the given bitsum protocol to bound the total error of any output KDE estimate.

### A.1.1 DISCRETIZATION

We will index the users in the protocol by $u = 1, \ldots, n$. Fix $i \in [I]$. Let $(f_i, g_i)$ be the pair sampled from $\mathcal{Q}$ in the global initialization step of Algorithm 1, and recall that $f_i, g_i : \mathbb{R}^d \to [-R, R]^Q$. Consider a user $u \in [n]$ with input $x_u \in \mathbb{R}^d$. In the randomizer of Algorithm 1, for every $j \in [Q]$, the user samples $b_{ij} \sim \text{Bernoulli}((f_i(x_u)_j + R)/2R)$ using private randomness, independently of the other users and of the other coordinates. To refer to local samples of different users, in this proof we will denote $b_{ij}$ by $b_{ij}^{(u)}$.

Define $\bar{f}_i^{(u)} \in \{-R, R\}^Q$ by letting $\bar{f}_{ij}^{(u)} = (2b_{ij}^{(u)} - 1)R$ for every $j$.

**Claim A.2.** *For every $i \in [I]$, $u \in [n]$ and $y \in \mathbb{R}^d$,*

$$\left| \mathbb{E}_{(f_i, g_i), \{b_{ij}^{(u)}\}_{j=1}^Q} \left[ (\bar{f}_i^{(u)})^T g_i(y) \right] - \mathbf{k}(x, y) \right| \le \beta,$$

*where the expectation is over both the sampling of $(f_i, g_i) \sim \mathcal{Q}$ in the global initialization part and the sampling of $\{b_{ij}^{(u)} : j \in [Q]\}$ in the randomizer part of Algorithm 1.*

*Proof.* It is immediate to check that $\mathbb{E}_{b_{ij}^{(u)}}[\bar{f}_{ij}^{(u)} \mid f_i] = f_i(x_u)_j$ for every $j$, hence,

$$\mathbb{E}_{(f_i, g_i), \{b_{ij}^{(u)}\}} \left[ (\bar{f}_i^{(u)})^T g_i(y) \right] = \mathbb{E}_{(f_i, g_i)} \left[ \mathbb{E}_{\{b_{ij}^{(u)}\}} \left[ (\bar{f}_i^{(u)})^T g_i(y) \mid (f_i, g_i) \right] \right]$$
$$= \mathbb{E}_{(f_i, g_i)} \left[ f_i(x)^T g_i(y) \right],$$

and the claim follows from the LSQ property (Definition 2.1). $\square$

### A.1.2 INSTANCES OF THE BITSUM PROTOCOL

Fix $(i, j) \in [I] \times [Q]$. Let $\bar{F}_{ij} = \sum_{u=1}^n \bar{f}_{ij}^{(u)}$ and $B_{ij} = \sum_{u=1}^n b_{ij}^{(u)}$. The shuffled DP protocol in Algorithm 1 executes an independent instance of the given shuffled DP bitsum protocol $\Pi$ to estimate $B_{ij}$, and this estimate is denoted by $\widetilde{B}_{ij}$ in the analyzer in Algorithm 1. We will denote this instance of $\Pi$ by $\Pi_{ij}$. Recall that $\Pi$ is an unbiased bitsum protocol and has RMSE $\mathcal{E}_\Pi$. Since $B_{ij}$ itself is a random variable determined by the sampling of $(f_i, g_i) \sim \mathcal{Q}$ and on the local randomized rounding by the users, conditioning on these, we have

$$\mathbb{E}_{\Pi_{ij}}[\widetilde{B}_{ij} - B_{ij} \mid (f_i, g_i), B_{ij}] = 0 \quad \text{and} \quad \mathbb{E}_{\Pi_{ij}}[|\widetilde{B}_{ij} - B_{ij}|^2 \mid (f_i, g_i), B_{ij}] = \mathcal{E}_\Pi^2.$$

The analyzer computes and publishes $\widetilde{F}_{ij} = (2\widetilde{B}_{ij} - n)R$. Considering this as an estimate of $\bar{F}_{ij}$, we denote

$$E_{ij} := \widetilde{F}_{ij} - \bar{F}_{ij}.$$

**Claim A.3.** $\{E_{ij} \mid (f_i, g_i), B_{ij}\}_{i,j}$ *are independent random variables, and each satisfies*

$$\mathbb{E}_{\Pi_{ij}}\left[E_{ij} \mid (f_i, g_i), B_{ij}\right] = 0 \quad and \quad \mathbb{E}_{\Pi_{ij}}\left[|E_{ij}|^2 \mid (f_i, g_i), B_{ij}\right] = (2R \cdot \mathcal{E}_\Pi)^2.$$

*Proof.* Probabilistic independence holds since we use independent randomness in the instance $\Pi_{ij}$ of $\Pi$ for different pairs $i, j$. For the first and second moments, recall that $\bar{f}_{ij}^{(u)} = (2b_{ij}^{(u)} - 1)R$ for every user $u$, which implies $\bar{F}_{ij} = (2B_{ij} - n)R$ when summing over the users. Also recall from above that $\widetilde{F}_{ij} = (2\widetilde{B}_{ij} - n)R$. Thus,

$$\begin{aligned}
\mathbb{E}_{\Pi_{ij}}\left[E_{ij} \mid (f_i, g_i), B_{ij}\right] &= \mathbb{E}_{\Pi_{ij}}\left[\widetilde{F}_{ij} - \bar{F}_{ij} \mid (f_i, g_i), B_{ij}\right] \\
&= \mathbb{E}_{\Pi_{ij}}\left[(2\widetilde{B}_{ij} - n)R - (2B_{ij} - n)R \mid (f_i, g_i), B_{ij}\right] \\
&= 2R \cdot \mathbb{E}_{\Pi_{ij}}\left[\widetilde{B}_{ij} - B_{ij} \mid (f_i, g_i), B_{ij}\right] \\
&= 0,
\end{aligned}$$

and

$$\begin{aligned}
\mathbb{E}_{\Pi_{ij}}\left[|E_{ij}|^2 \mid (f_i, g_i), B_{ij}\right] &= \mathbb{E}_{\Pi_{ij}}\left[\left|\widetilde{F}_{ij} - \bar{F}_{ij}\right|^2 \mid (f_i, g_i), B_{ij}\right] \\
&= \mathbb{E}_{\Pi_{ij}}\left[\left|(2\widetilde{B}_{ij} - n)R - (2B_{ij} - n)R\right|^2 \mid (f_i, g_i), B_{ij}\right] \\
&= (2R)^2 \cdot \mathbb{E}_{\Pi_{ij}}\left[\left|\widetilde{B}_{ij} - B_{ij}\right|^2 \mid (f_i, g_i), B_{ij}\right] \\
&= (2R \cdot \mathcal{E}_\Pi)^2.
\end{aligned}$$

$\square$

### A.1.3  BOUNDING THE SUPRMSE

To bound the supRMSE of Algorithm 1, fix $y \in \mathbb{R}^d$. The KDE query part of the protocol uses the analyzer's published output to estimate $KDE_X(y)$ by $\frac{1}{nI} \sum_{i=1}^{I} \sum_{j=1}^{Q} \widetilde{F}_{ij} g_i(y)_j$. We now bound the RMSE of this estimate. Substituting $E_{ij} := \widetilde{F}_{ij} - \bar{F}_{ij}$, we have

$$\begin{aligned}
&\mathbb{E}\left[\left|KDE_X(y) - \frac{1}{nI} \sum_{i=1}^{I} \sum_{j=1}^{Q} \widetilde{F}_{ij} g_i(y)_j\right|^2\right] \\
&= \mathbb{E}\left[\left|KDE_X(y) - \frac{1}{nI} \sum_{i=1}^{I} \sum_{j=1}^{Q} \bar{F}_{ij} g_i(y)_j + \frac{1}{nI} \sum_{i=1}^{I} \sum_{j=1}^{Q} E_{ij} g_i(y)_j\right|^2\right] \\
&\leq 2\mathbb{E}\left[\left|KDE_X(y) - \frac{1}{nI} \sum_{i=1}^{I} \sum_{j=1}^{Q} \bar{F}_{ij} g_i(y)_j\right|^2\right] + 2\mathbb{E}\left[\left|\frac{1}{nI} \sum_{i=1}^{I} \sum_{j=1}^{Q} E_{ij} g_i(y)_j\right|^2\right]. \quad (2)
\end{aligned}$$

We handle the two summands in turn.

### A.1.4 FIRST SUMMAND: DISCRETIZED LSQ APPROXIMATION ERROR

For every $i$, let $\bar{F}_i \in \mathbb{R}^Q$ denote the vector with coordinates $\bar{F}_{ij}$. Recalling that $\bar{F}_{ij} = \sum_{u=1}^n \bar{f}_{ij}^{(u)}$, we have $\bar{F}_i = \sum_{u=1}^n \bar{f}_i^{(u)}$. We can thus write,

$$\mathbb{E}\left[\left\|KDE_X(y) - \frac{1}{nI}\sum_{i=1}^I\sum_{j=1}^Q \bar{F}_{ij}g_i(y)_j\right\|^2\right] = \mathbb{E}\left[\left\|KDE_X(y) - \frac{1}{nI}\sum_{i=1}^I \bar{F}_i^T g_i(y)\right\|^2\right]$$

$$= \frac{1}{I^2}\mathbb{E}\left[\left\|\sum_{i=1}^I\left(KDE_X(y) - \frac{1}{n}\bar{F}_i^T g_i(y)\right)\right\|^2\right]. \quad (3)$$

Denote the random variables,

$$Z_i := KDE_X(y) - \frac{1}{n}\bar{F}_i^T g_i(y),$$

and for every $u \in [n]$,

$$Z_{i,u} := \mathbf{k}(x_u, y) - (\bar{f}_i^{(u)})^T g_i(y).$$

Observe that $Z_i = \frac{1}{n}\sum_{u=1}^n Z_{i,u}$, and that the rightmost side of Equation (3) is $\frac{1}{I^2}\mathbb{E}[|\sum_{i=1}^I Z_i|^2]$. Due to the probabilistic independence of samples for different values $i \in [I]$, we can expand this as

$$\mathbb{E}\left[\left\|\sum_{i=1}^I Z_i\right\|^2\right] = \left|\sum_{i=1}^I \mathbb{E}[Z_i^2] + \sum_{i=1}^I\sum_{i'\neq i}\mathbb{E}[Z_i]\cdot\mathbb{E}[Z_{i'}]\right|$$

$$\leq \sum_{i=1}^I \mathbb{E}[Z_i^2] + \sum_{i=1}^I\sum_{i'\neq i}|\mathbb{E}[Z_i]|\cdot|\mathbb{E}[Z_{i'}]|. \quad (4)$$

**Claim A.4.** *For every $i$ we have $|\mathbb{E}[Z_i]| \leq \beta$ and $\mathbb{E}[Z_i^2] \leq (\beta + 2R^2S)^2$.*

*Proof.* For the first bound in the claim, observe that Claim A.2 can be rewritten as $|\mathbb{E}[Z_{i,u}]| \leq \beta$ for every $i, u$. Since $Z_i = \frac{1}{n}\sum_{u=1}^n Z_{i,u}$, we get $|\mathbb{E}[Z_i]| \leq \frac{1}{n}\sum_{u=1}^n \mathbb{E}|Z_{i,u}| \leq \beta$.

For the second bound in the claim, recall that by Definition 2.1, for every supported function pair $(f, g)$ in the LSQ family $\mathcal{Q}$, and every $x, y \in \mathbb{R}^d$, we have that $f(x)$ and $g(y)$ have coordinates in $[-R, R]$, and have at most $S$ non-zero coordinates each. Thus, $|f(x)^T g(y)| \leq R^2S$. By recalling that $\bar{f}_i^{(u)}$ was generated from $f_i(x_u)$ by rounding its coordinates to $\{-R, R\}$, this implies in particular that $|(\bar{f}_i^{(u)})^T g_i(y)| \leq R^2S$ for every $u$. Moreover, since by Definition 2.1 we have $|\mathbf{k}(x, y) - \mathbb{E}_{(f,g)\sim\mathcal{Q}}[f(x)^T g(y)]| \leq \beta$, this also implies that $|\mathbf{k}(x, y)| \leq R^2S + \beta$. Therefore, unconditionally,

$$|Z_i| \leq \frac{1}{n}\sum_{u=1}^n |Z_{i,u}|$$

$$= \sum_{u=1}^n \left|\mathbf{k}(x_u, y) - (\bar{f}_i^{(u)})^T g_i(y)\right|$$

$$\leq \frac{1}{n}\sum_{u=1}^n \left(|\mathbf{k}(x_u, y)| + \left|(\bar{f}_i^{(u)})^T g_i(y)\right|\right)$$

$$\leq \beta + 2R^2S,$$

which implies in particular $\mathbb{E}[Z_i^2] \leq (\beta + 2R^2S)^2$. $\qquad\square$

We now have,

$$\mathbb{E}\left[\left|\left|KDE_X(y) - \frac{1}{nI}\sum_{i=1}^{I}\sum_{j=1}^{Q}\bar{F}_{ij}g_i(y)_j\right|\right|^2\right]$$

$$= \frac{1}{I^2}\mathbb{E}\left[\left|\left|\sum_{i=1}^{I}\left(KDE_X(y) - \frac{1}{n}\bar{F}_i^T g_i(y)\right)\right|\right|^2\right] \qquad \text{Equation (3)}$$

$$= \frac{1}{I^2}\mathbb{E}\left[\left|\left|\sum_{i=1}^{I}Z_i\right|\right|^2\right] \qquad \text{definition of } Z_i$$

$$\leq \frac{1}{I^2}\left(\sum_{i=1}^{I}\mathbb{E}[Z_i^2] + \sum_{i=1}^{I}\sum_{i'\neq i}|\mathbb{E}[Z_i]|\cdot|\mathbb{E}[Z_{i'}]|.\right) \qquad \text{Equation (4)}$$

$$\leq \frac{1}{I^2}\left(I(\beta + 2R^2S)^2 + I(I-1)\beta^2\right) \qquad \text{Claim A.4}$$

$$\leq \frac{8R^4S^2}{I} + 2\beta^2.$$

This is our bound for the first summand in Equation (2).

### A.1.5 SECOND SUMMAND: TOTAL BITSUM PROTOCOL ERROR

For $i \in [I]$, let $Y_i$ be the random variable

$$Y_i = \sum_{j=1}^{Q}E_{ij}g_i(y)_j.$$

Note that the second summand in Equation (2) equals $2(\frac{1}{nI})^2\mathbb{E}[(\sum_{i=1}^{I}Y_i)^2]$.

By Claim A.3, $\{E_{ij} \mid (f_i, g_i), B_{ij}\}_{i,j}$ are independent random variables. Each $Y_i$, when conditioned on $(f_i, g_i), \{B_{ij}\}_{j\in[Q]}$, is a linear combination of a subset of these random variables and the subsets are disjoint for $i \neq i'$, hence $\{Y_i \mid (f_i, g_i), \{B_{ij}\}_{j\in[Q]}\}_{i\in[I]}$ are also independent random variables. Furthermore, for every $i \in [I]$ we have

$$\mathbb{E}_{\{\Pi_{ij}\}_{j\in[Q]}}\left[Y_i \mid (f_i, g_i), \{B_{ij}\}_{j\in[Q]}\right] = \mathbb{E}_{\{\Pi_{ij}\}_{j\in[Q]}}\left[\sum_{j=1}^{Q}E_{ij}g_i(y)_j \mid (f_i, g_i), \{B_{ij}\}_{j\in[Q]}\right]$$

$$= \sum_{j=1}^{Q}g_i(y)_j\mathbb{E}_{\Pi_{ij}}\left[E_{ij} \mid (f_i, g_i), B_{ij}\right]$$

$$= 0,$$

and

$$\mathbb{E}_{\{\Pi_{ij}\}_{j\in[Q]}}\left[Y_i^2 \mid (f_i, g_i), \{B_{ij}\}_{j\in[Q]}\right] = \mathbb{E}_{\{\Pi_{ij}\}_{j\in[Q]}}\left[\left(\sum_{j=1}^{Q}E_{ij}g_i(y)_j\right)^2 \mid (f_i, g_i), \{B_{ij}\}_{j\in[Q]}\right]$$

$$= \sum_{j=1}^{Q}(g_i(y)_j)^2\mathbb{E}_{\Pi_{ij}}\left[E_{ij}^2 \mid (f_i, g_i), B_{ij}\right]$$

$$= \sum_{j=1}^{Q}(g_i(y)_j)^2(2R\mathcal{E}_\Pi)^2,$$

having used $\mathbb{E}_{\Pi_{ij}}\left[E_{ij} \mid (f_i, g_i), B_{ij}\right] = 0$ and $\mathbb{E}_{\Pi_{ij}}\left[\left|E_{ij}\right|^2 \mid (f_i, g_i), B_{ij}\right] = (2R \cdot \mathcal{E}_\Pi)^2$ from Claim A.3. Since by Definition 2.1 $g_i(y)$ has at most $S$ non-zero entries and each is bounded in absolute value by $R$,

$$\mathbb{E}_{\{\Pi_{ij}\}_{j\in[Q]}}\left[|Y_i|^2 \mid (f_i, g_i), \{B_{ij}\}_{j\in[Q]}\right] \le 4SR^4 \cdot \mathcal{E}_\Pi^2.$$

Therefore,

$$\mathbb{E}_{\{\Pi_{ij}\}_{i\in[I],j\in[Q]}}\left[\left(\sum_{i=1}^I Y_i\right)^2 \mid \{(f_i, g_i), B_{ij}\}_{i\in[I],j\in[Q]}\right]$$
$$= \sum_{i=1}^I \mathbb{E}_{\{\Pi_{ij}\}_{j\in[Q]}}\left[Y_i^2 \mid (f_i, g_i), \{B_{ij}\}_{j\in[Q]}\right]$$
$$\le I \cdot 4SR^4 \cdot \mathcal{E}_\Pi^2.$$

Now we can bound the second summand in Equation (2) as

$$\frac{2}{n^2 I^2}\mathbb{E}_{\{(f_i,g_i),B_{ij},\Pi_{ij}\}_{i\in[I],j\in[Q]}}\left[\left(\sum_{i=1}^I Y_i\right)^2\right]$$
$$= \frac{2}{n^2 I^2}\mathbb{E}_{\{(f_i,g_i),B_{ij}\}_{i\in[I],j\in[Q]}}\left[\mathbb{E}_{\{\Pi_{ij}\}_{i\in[I],j\in[Q]}}\left[\left(\sum_{i=1}^I Y_i\right)^2 \mid \{(f_i,g_i),B_{ij}\}_{i\in[I],j\in[Q]}\right]\right]$$
$$\le \frac{8SR^4\mathcal{E}_\Pi^2}{n^2 I}.$$

### A.1.6 FINISHING THE PROOF OF THEOREM 3.2

**Accuracy:** By putting together the bounds on both summands in Equation (2), we get that the RMSE of estimating $KDE_X(y)$ is at most $\sqrt{4\beta^2 + \frac{16R^4 S}{I}\left(S + \frac{\mathcal{E}_\Pi^2}{n^2}\right)}$. Since this holds for every $y \in \mathbb{R}^d$, this is a bound on the supRMSE.

**Privacy:** for every $i \in [I]$ and $j \in [Q]$, let $O_{ij}$ denote the output of the shuffler in protocol instance $\Pi_{ij}$. The fact that $\Pi_{ij}$ is an instance of the $(\varepsilon_0, \delta_0)$-DP protocol $\Pi$ means (by the definition of the shuffled DP model) that $O_{ij}$ is $(\varepsilon_0, \delta_0)$-DP w.r.t. the collection of user inputs.

First, fix $i \in [I]$. Recall the sparsity property of LSQ (Definition 2.1), namely that each $f_i$ has at most $S$ non-zero entries per user. This means that if the input of one user is omitted from the dataset, the inputs of at most $S$ of the $Q$ protocols $\{\Pi_{ij}\}_{j=1}^Q$ are changed. Since these protocol instances use independent randomness, then by standard ("basic") DP composition arguments (Dwork et al., 2014), the collection $\{O_{ij}\}_{j=1}^Q$ is $(\varepsilon_0 S, \delta_0 S)$-DP. In other words, the protocol $\Psi_i$ obtained by composing the protocols $\{\Pi_{ij}\}_{j=1}^Q$ is $(\varepsilon_0 S, \delta_0 S)$-DP in the shuffled model (see Cheu et al. (2019) for the definition of protocol composition in the shuffled DP model).

Now, by "advanced" composition for shuffled DP protocols (Lemma 3.6 in Cheu et al. (2019)) over the $I$ protocols $\{\Psi_i\}$, we get that the collection of shuffler outputs $\{O_{ij} : (i, j) \in [I] \times [Q]\}$ is $(\epsilon, \delta)$-DP, with $\epsilon, \delta$ as stated in Theorem 3.2. Since the analyzer in Algorithm 1 is a post-processing of these shuffler outputs, the protocol in Algorithm 1 is $(\epsilon, \delta)$-DP in the shuffled model.

**Efficiency:** The computational parameters of Algorithm 1 is straightforward to calculate from those of the bitsum protocol $\Pi$ and the LSQ family $\mathcal{Q}$: the global initialization samples $I$ pairs $(f_i, g_i)$ from $\mathcal{Q}$; each user evaluates $f_i(x)$ on her input $x$ for every $i$; the users, the shuffler and the analyzer perform $IQ$ instances of $\Pi$, thus incurring $IQ$ times its computational and communication cost; the the final KDE evaluation part evaluates $g_i$ on the query $y$ for every $i \in [I]$. $\qquad\square$

### A.1.7 VARIANTS OF THEOREM 3.2

The foregoing proof of Theorem 3.2 can be adapted in various ways to accommodate bitsum protocols with different properties than those stated in the theorem. For example,

---

**Algorithm 2:** Shuffled DP Gaussian KDE protocol, based on either RR or 3NB bitsum protocol

---

**Global initialization**  *// all data here is public*
  **input:** integer $I > 0$;
  parameters for RR bitsum: $p_{RR} \in (0,1)$;
  parameters for 3NB bitsum: $r, r', p, p' > 0$

  **for** $i = 1, \ldots, I$ **do**
    *// i.i.d. samples using shared/public*
      *randomness:*
    $\omega_i \sim N(0, \mathbb{I}_d)$    *// d-dim normal r.v.*
    $\beta_i \sim \text{Uniform}[0, 2\pi)$
  **publish:** $\omega_i$ and $\beta_i$ for all $i$

**Randomizer**  *// each user runs this locally with*
 *private randomness*
  **input:** private data point $x \in \mathbb{R}^d$
  **for** $i = 1, \ldots, I$ **do**
    $\varphi_i \leftarrow \cos(\sqrt{2}\omega_i^T x + \beta_i)$
    $b_i \sim \text{Bernoulli}((1 + \varphi_i)/2)$
    **if** *bitsum protocol is RR* **then**
      $b_i \leftarrow$ flip with probability $p_{RR}$
      send $(b_i, i)$ to the shuffler

    **if** *bitsum protocol is 3NB* **then**
      $\psi_1 \sim \text{NegativeBinomial}(r, p)$
      $\psi_2 \sim \text{NegativeBinomial}(r, p)$
      $\psi_3 \sim \text{NegativeBinomial}(r', p')$
      **for** $j = 1, \ldots, b_i + \psi_1 + \psi_3$ **do**
        send $(1, i)$ to the shuffler
      **for** $j = 1, \ldots, \psi_2 + \psi_3$ **do**
        send $(-1, i)$ to the shuffler

**Analyzer**  *// runs after the shuffler; analyzer is*
 *the same for both RR and 3NB bitsum*
 *protocols*
  **input:** shuffled sequence of messages $\widetilde{\Gamma}$
      from $n$ users
  **for** $i = 1, \ldots, I$ **do**
    $\widetilde{B}_i \leftarrow 0$
  **for** *message* $(\gamma, i)$ *in* $\widetilde{\Gamma}$ **do**
    $\widetilde{B}_i \leftarrow \widetilde{B}_i + \gamma$
  **for** $i = 1, \ldots, I$ **do**
    $\widetilde{F}_i \leftarrow 2\widetilde{B}_i - n$
  **publish:** $\widetilde{F}_i$ for all $i$

**KDE Query**  *// runs on the analyzer's*
 *published output arbitrarily many times*
  **input:** query point $y \in \mathbb{R}^d$
  **return:** $\frac{2}{nI} \sum_{i=1}^{I} \widetilde{F}_i \cdot \cos(\sqrt{2}\omega_i^T y + \beta_i)$

---

- If the accuracy guarantee of $\Pi$ is given in terms of absolute error rather than RMSE (as in Cheu et al. (2019)), the proof can be repeated with bounding the supremum absolute error of Algorithm 1 instead of its supRMSE (this yields a very similar and somewhat simpler version of the proof given above).
- If $\Pi$ has a pure DP guarantee, the advanced composition step in the privacy analysis from Appendix A.1.6 can be replaced by standard pure composition, resulting in $\varepsilon = IS\varepsilon_0$ (compare this to $\varepsilon \sim \sqrt{I}S\varepsilon_0$ in Theorem 3.2) and $\delta = 0$. In the analogous instantiation of Theorem 3.3, the lower bound on the error $\alpha$ changes from $\sqrt{\log(1/\delta)}/(\varepsilon n)$ to $1/\sqrt{\varepsilon n}$.
- If $\Pi$ is not unbiased, the proof (specifically Appendix A.1.5) can be slightly modified to accommodate its bias, resulting in a corresponding term in the final supRMSE bound.

## A.2  Proof of Theorem 3.3

We restate Theorem 3.3 and prove it as a corollary of Theorem 3.2. The corresponding protocol for Gaussian KDE is Algorithm 2 with the choice of 3NB as the bitsum protocol.[4]

**Theorem A.5** (Theorem 3.3, restated)**.** *There are constants $C, C' > 0$ such that the following holds. Let $\delta \in (0,1)$ and $\varepsilon \leq C \log(1/\delta)$. For every $\alpha \geq C'\sqrt{\log(1/\delta)}/(\varepsilon n)$, there is an $(\varepsilon, \delta)$-DP Gaussian KDE protocol in the shuffled DP model (under the communication-threat model) with $n$ users and inputs from $\mathbb{R}^d$, which has: supRMSE $\alpha$, user running time $\min(O(d/\alpha^2), \tilde{O}(d + 1/\alpha^4))$, expected communication of $\tilde{O}(1/\alpha^2)$ bits per user, expected analyzer running time $O(n/\alpha^2)$, KDE query time $\min(O(d/\alpha^2), \tilde{O}(d + 1/\alpha^4))$, and optimal bit-width 1.*

---

[4]For completeness, Algorithm 2 also specifies how to use RR as the bitsum protocol. The flip probability $p_{RR}$ should be set according to Lemma 4.8 in Chen et al. (2020a).

*Proof.* Recall that the Gaussian kernel has an LSQ family with $\beta = 0$, $Q = S = 1$, $R = \sqrt{2}$ by random Fourier features. For clarity, we mostly suppress constants in this proof. For the given $\varepsilon, \delta, \alpha$ in Theorem 3.3, we set the parameters in Theorem 3.2 as follows:

$$I = \lceil \frac{1}{\alpha^2} \rceil \;\; ; \;\; \varepsilon_0 = \frac{\varepsilon}{\sqrt{I \log(1/\delta)}} \;\; ; \;\; \delta_0 = \frac{\delta}{2I} \;\; ; \;\; \delta' = \delta/2.$$

**Privacy:** It can be easily checked that plugging the above setting of parameters into the composed privacy parameters in Theorem 3.2 yields an $(\varepsilon, \delta)$-DP guarantee in the shuffled model, provided that the given bitsum protocol $\Pi$ is $(\varepsilon_0, \delta_0)$-DP.

To this end, we use the 3NB bitsum protocol from Ghazi et al. (2020b) as $\Pi$, since it has near-optimal accuracy with low communication overhead. 3NB has four parameters $r, r', p, p'$ (see Algorithm 2) that Ghazi et al. (2020b) show how to set to ensure the protocol is $(\varepsilon_0, \delta_0)$-DP in the shuffled model. Namely, they prove that setting $r = 1/n$, $p = e^{-0.99\varepsilon_0}$, $r' = 3(1 + \log(2e^{0.99\varepsilon_0}/\delta_0))$, $p' = e^{-\Theta(1)\cdot\varepsilon_0/(\varepsilon_0+\log(1/\delta_0))}$ guarantees 3NB is $(O(\varepsilon_0), O(\delta_0))$-DP, and the constants can be scaled so it is $(\varepsilon_0, \delta_0)$-DP.

**Accuracy:** The 3NB protocol is unbiased and has RMSE $\Theta(1/\varepsilon_0) = \Theta(\sqrt{\log(1/\delta)}/(\alpha\varepsilon))$. Plugging this into the supRMSE in Theorem 3.2, we get supRMSE $O\left(\sqrt{\alpha^2(1 + \log(1/\delta)/(\alpha\varepsilon n)^2)}\right)$ in Algorithm 2. By the bound on $\alpha$ in the statement of Theorem 3.3, this supRMSE is at most $O(\alpha)$, and we can scale the constants to get supRMSE $\alpha$.

**Efficiency:** We recall that in the 3NB protocol from Ghazi et al. (2020b), each user runs in $O(1)$ time and sends an expected nunber of $1 + o(1)$ messages of $O(1)$ bits each, which the analyzer iterates over in time $O(n)$. Since we have $I = O(1/\alpha^2)$ instances of this protocol, each message needs to include $O(\log(1/\alpha))$ additional bits to identify which protocol instance it belongs to, yielding $O(\log(1/\alpha)/\alpha^2)$ expected bits of communication per user, and expected analyzer running time $O(n/\alpha^2)$.

Each user also needs to compute the inner product $\omega_i^T x$ for every $i \in [I]$. Similarly, the KDE query algorithm needs to compute $\omega_i^T y$ for every $i \in [I]$. This takes time $O(d)$ per inner product, for a total of $O(dI) = O(d/\alpha^2)$ time. If $d \gg 1/\alpha^2$, this time bound can be improved by using the faster preprocessing result of Backurs et al. (2024), who showed that one can first do a random projection of $x$ (for each user input $x$) and $y$ (for each KDE query $y$) onto $O(\log^2(1/\alpha)/\alpha^2)$ dimensions, and thus only distort the final DP KDE error up to a multiplicative constant (that can again be scaled). As shown by Backurs et al. (2024), the random projection can be done in time $\tilde{O}(d + 1/\alpha^2)$ by the fast Johnson-Lindenstrauss transform (Ailon & Chazelle, 2009), and then each of the $I = O(1/\alpha^2)$ inner products takes time $\tilde{O}(1/\alpha^2)$, for a total of $\tilde{O}(d+1/\alpha^4)$ time per user and per KDE query. $\square$

## A.3 INNER PRODUCT LSQ

The inner product kernel $\mathbf{k}(x, y) = x^T y$ is trivially $(d, 1, d)$-LSQable for unit length embeddings, by letting the LSQ family include a single pair of functions $(f, g)$ such that both are the identity over $\mathbb{R}^d$. We now observe it is also $(1, \sqrt{d}, 1)$-LSQable. This allows better control over the privacy parameters and computational cost of the protocol in Theorem 3.2, since they depend on the parameters $S$ and $Q$ (respectively) of the $(Q, R, S)$-LSQ family.

To sample a pair $(f, g) \sim \mathcal{Q}$, we sample a vector $(\sigma_1, \ldots, \sigma_d) \in \{-1, 1\}^d$ of i.i.d. uniformly random signs, and let both $f$ and $g$ be the function $\mathbb{R}^d \to \mathbb{R}$ that maps $x = (x_1, \ldots, x_i)$ to $\sum_{i=1}^d \sigma_i x_i$. It is straightforward to check that $\mathbb{E}[f(x)^T g(y)] = x^T y$ for every $x, y \in \mathbb{R}^d$. To determine the upper bound $R$ on the only coordinate of $f(x)$, we observe, $|f(x)| = |\sum_{i=1}^d \sigma_i x_i| \leq \|x\|_1 \leq \sqrt{d}\|x\|_2 = \sqrt{d}$, since $x$ is unit length (in Euclidean norm).

# B    MORE ON SHUFFLED DP SUMMATION

## B.1    BITSUM PROTOCOLS

In this section we describe some common techniques behind shuffled DP bitsum protocols, including the ones we use in our experiments (RR, 3NB and Pure).

One bitsum protocol is the classical *randomized response* (RR) (Warner, 1965): each user $i$ locally flips her bit $b_i$ with some probability, and sends the resulting bit to the shuffler. The analyzer received the anonymized received bits from the shuffler, and simply releases their sum. While originally introduced for local DP, Cheu et al. (2019) showed that in the shuffled DP model, the flip probability can be significantly smaller, leading to much better accuracy.

Another popular technique for shuffled DP bitsums, which is the one underlying 3NB and Pure, is *noise divisibility* (Goryczka & Xiong, 2015; Balle et al., 2019b; 2020a; Ghazi et al., 2020b;a; 2021b; Kairouz et al., 2021a). Each user $i$ locally adds noise $\nu_i$, sampled from a distribution carefully chosen so that the aggregate noise $\sum_i \nu_i$ from all users, after shuffling, is distributed in a way that ensures central DP. Thus, the shuffled DP protocol simulates central DP, by having each user contribute a piece of the total "divisible" requisite noise.

To be concrete, we describe the single-distribution protocol from Ghazi et al. (2020b). In this protocol, each user $i$ samples a non-negative integer noise random variable $\nu_i$, and sends to the shuffler a stream of $b_i + \nu_i$ identical content-less messages (where $b_i$ is user $i$'s private bit). The analyzer receives the unified streams of messages from all users after shuffling. Since the messages are identical and are now stripped of both content and sender identities, the only information they convey is their count $\sum_i (b_i + \nu_i)$, which is released as the bitsum estimate. This equals the true bitsum $\sum_i b_i$ plus a total noise of $\sum_i \nu_i$. Thus, to ensure shuffled DP, it suffices for the $\nu_i$s to be such that their sum $\sum_i \nu_i$ is distributed in a way that ensures central DP for $\sum_i b_i$. Ghazi et al. (2020b) show this can be achieved by either a Poisson or a negative binomial distribution. The 3NB and Pure bitsum protocols are more involved applications of this basic technique, designed to achieve better accuracy, lower communication cost, and (in the case of Pure) a pure DP guarantee.

## B.2    BITSUMS VS. REAL SUMS

As mentioned in Section 2.4, there is also ample work on shuffled DP protocol for real number summation (abbrev. *realsum*), where each user holds an input number in a bounded range (say, $[-1, 1]$). In this appendix we expand on the choice to base our approach on bitsum rather than realsum protocols. The answer has two parts: (1) why there is little potential gain in real summation, (2) why there is substantial advatnage in bit summation.

**Little gain in realsums.** Ostensibly, Algorithm 1 and Theorem 3.2 could have used realsum instead of bitsum protocols, obviating the need to discretizate the LSQ coordinates with randomized rounding. When summing $\ell$ real numbers in $[-1, 1]$, discretization with randomized rounding generally leads to a Hoeffding-like error of order $\sqrt{\ell}$, which our protocol incurs. There are parameter regimes where shuffled DP realsum protocols are more accurate than bitsum protocols, avoiding this Hoeffding-like error, and thus it may seem like an avenue to improve Theorem 3.2.

However, this is in fact not the case. In our protocol, summation serves as a subroutine. The true sum is not the target quantity; rather, the true sum is a random variable (sampled according to the LSQ family), which only approximates the target quantity against which error is measured (the true KDE). This LSQ approximation already incurs the Hoeffding-like error (it is "built-into" LSQ). Thus, if the discretized bitsums were to be replaced with a shuffled DP realsum protocol, the final KDE error would still be dominated by the Hoeffding-like error, and any improvement would be restricted to low-order terms. Improving the final KDE error asymptotically, if this is indeed possible, would require a different approach than LSQ to private KDE, and we are currently not aware of a way to improve the KDE error in the shuffled DP model.

**Advangate of bitsums.** At the same time, discretization has its own important advantages in shuffled DP and distributed learning. This was discussed in Section 3.1. To recap, in practical applications of shuffled DP, numerical values need to be discretized, and their bit-width bounded, in order to properly control their accuracy and communication cost. This was among the main motivations of

Kairouz et al. (2021a) in developing their Distributed Discrete Gaussian (DDG) protocol for shuffled DP realsum, rather than using prior protocols for this task. The DDG facilitates bounding the bit-width (though it is not as low as 1), and its error analysis accounts for discretization errors. Our approach, which does not need to solve generic real summation, but only the specific case of KDE, attains the optimal bit-width of 1 by using binary discretization followed by bit summation, and our error analysis too accounts for the discretization error.

## C  FULL EXPERIMENTAL RESULTS

**Ablation: Local DP.**    Figure 4 displays a comparison of our shuffled DP method with local DP. The local DP baseline is obtained by taking our private KDE protocol (Theorem 3.2), and replacing the shuffled DP bitsum with classical randomized response, which satisfies local DP. For the most direct comparison, the local DP plots are displayed compared to the RR plots (i.e., the leftmost column) in Figure 1.

We recall that the difference between the methods is that in classical local DP RR (the dashed lines in the plot) (Warner, 1965), each user locally flips her bit with probability that depends on the desired privacy parameter $\varepsilon$, but is independent of the overall number of user in the protocol (local DP RR makes no assumptions on other participating users). In contrast, in shuffled DP RR (the solid plots) (Cheu et al., 2019), each user flips her bit with probability that depends both on $\varepsilon$ and on the total number of participating user $n$; the larger $n$ is, the smaller the flip probability needs to be, since in shuffled DP, the user assumes her bit would also be anonymized and "hidden" among the bits received from the other $n - 1$ users.

The results in Figure 4 show that as expected, shuffled DP attains considerably higher downstream accuracy than local DP.

**Ablation: Effect of $\varepsilon_{\mathrm{lbl}}$.**    Tables 2 to 5 show the effect of varying $\varepsilon_{\mathrm{lbl}}$ on various settings on the four datasets, respectively.

**Additional parameter settings.**    Figures 5 to 8 display private classification accuracy results for $\varepsilon_{\mathrm{lbl}} = 10, 7, 5, 3$ respectively (Figure 7 repeats Figure 1 from the main paper for convenience).

Tables 6 to 8 present private class decoding results with $\varepsilon_{\mathrm{lbl}} = 5$ and $\varepsilon \approx 5.7, 4.4, 3.2$ respectively (Table 8 is the full version of Table 1 from the main paper).

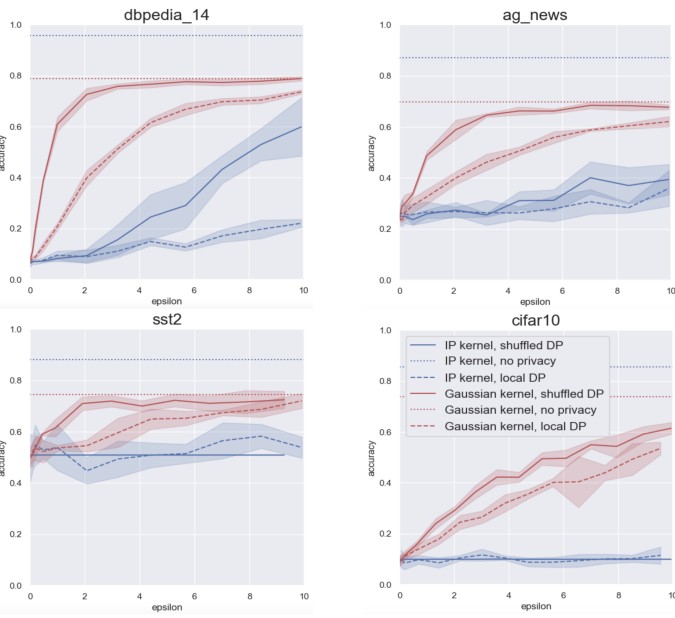

Figure 4: Classification accuracy comparison with a local DP baseline (overlaid on the shuffled DP RR plots with $\varepsilon_{\mathrm{lbl}} = 5$, from the leftmost column in Figure 1).

Table 2: Effect of $\varepsilon_{lbl}$ on accuracy, DBPedia-14.

| Kernel | Bitsum | $\varepsilon$ | % Classification accuracy ($\pm$std) with $\varepsilon_{lbl}$: | | | | |
|---|---|---|---|---|---|---|---|
| | | | $\infty$ | 7 | 5 | 3 | 1 |
| Gaussian | RR | 2 | $75.5_{\pm1.1}$ | $76.1_{\pm1.8}$ | $73.6_{\pm1.1}$ | $57.1_{\pm2.6}$ | $15.2_{\pm1.0}$ |
| | RR | 4.5 | $79.7_{\pm1.3}$ | $79.3_{\pm1.9}$ | $76.1_{\pm0.9}$ | $61.9_{\pm2.5}$ | $15.9_{\pm0.7}$ |
| | RR | 7 | $79.5_{\pm1.5}$ | $79.1_{\pm0.8}$ | $77.4_{\pm1.7}$ | $63.6_{\pm1.5}$ | $17.6_{\pm0.7}$ |
| | 3NB | 2 | $79.2_{\pm1.6}$ | $79.6_{\pm1.6}$ | $79.4_{\pm0.8}$ | $65.0_{\pm1.0}$ | $15.8_{\pm1.0}$ |
| | 3NB | 4.5 | $80.3_{\pm1.5}$ | $78.4_{\pm0.9}$ | $76.9_{\pm1.1}$ | $64.7_{\pm1.3}$ | $15.9_{\pm1.4}$ |
| | 3NB | 7 | $80.0_{\pm0.8}$ | $79.1_{\pm1.0}$ | $77.5_{\pm0.8}$ | $65.2_{\pm2.1}$ | $17.0_{\pm1.9}$ |
| IP | RR | 2 | $27.0_{\pm0.8}$ | $27.3_{\pm4.9}$ | $23.0_{\pm1.8}$ | $15.4_{\pm1.8}$ | $7.8_{\pm1.2}$ |
| | RR | 4.5 | $50.5_{\pm3.5}$ | $51.3_{\pm1.6}$ | $46.2_{\pm3.1}$ | $30.0_{\pm2.8}$ | $10.7_{\pm2.0}$ |
| | RR | 7 | $65.4_{\pm2.1}$ | $66.7_{\pm2.0}$ | $63.0_{\pm1.6}$ | $45.2_{\pm3.3}$ | $12.0_{\pm1.8}$ |
| | 3NB | 2 | $92.7_{\pm0.1}$ | $92.8_{\pm0.2}$ | $92.5_{\pm0.2}$ | $91.0_{\pm0.4}$ | $58.9_{\pm1.4}$ |
| | 3NB | 4.5 | $93.1_{\pm0.1}$ | $93.1_{\pm0.0}$ | $93.1_{\pm0.1}$ | $92.5_{\pm0.3}$ | $72.2_{\pm1.2}$ |
| | 3NB | 7 | $93.1_{\pm0.2}$ | $93.1_{\pm0.2}$ | $93.1_{\pm0.1}$ | $92.6_{\pm0.2}$ | $72.9_{\pm1.7}$ |

Table 3: Effect of $\varepsilon_{lbl}$ on accuracy, AG News.

| Kernel | Bitsum | $\varepsilon$ | % Classification accuracy ($\pm$std) with $\varepsilon_{lbl}$: | | | | |
|---|---|---|---|---|---|---|---|
| | | | $\infty$ | 7 | 5 | 3 | 1 |
| Gaussian | RR | 2 | $60.5_{\pm3.6}$ | $60.4_{\pm2.4}$ | $61.7_{\pm0.8}$ | $57.2_{\pm2.4}$ | $36.3_{\pm2.2}$ |
| | RR | 4.5 | $66.8_{\pm1.1}$ | $66.7_{\pm1.7}$ | $66.8_{\pm1.4}$ | $61.8_{\pm2.8}$ | $37.1_{\pm1.0}$ |
| | RR | 7 | $67.7_{\pm0.8}$ | $67.9_{\pm1.6}$ | $67.9_{\pm1.4}$ | $62.9_{\pm2.7}$ | $40.7_{\pm1.6}$ |
| | 3NB | 2 | $68.2_{\pm1.2}$ | $69.5_{\pm1.4}$ | $67.0_{\pm1.0}$ | $63.4_{\pm2.3}$ | $41.7_{\pm0.9}$ |
| | 3NB | 4.5 | $68.7_{\pm1.1}$ | $69.1_{\pm1.2}$ | $68.5_{\pm1.1}$ | $63.3_{\pm2.5}$ | $41.0_{\pm2.6}$ |
| | 3NB | 7 | $67.8_{\pm2.2}$ | $68.2_{\pm0.7}$ | $68.7_{\pm1.3}$ | $62.8_{\pm1.7}$ | $40.5_{\pm2.0}$ |
| IP | RR | 2 | $30.3_{\pm5.8}$ | $33.5_{\pm1.7}$ | $35.7_{\pm2.7}$ | $31.1_{\pm2.0}$ | $25.4_{\pm3.0}$ |
| | RR | 4.5 | $45.5_{\pm3.7}$ | $41.4_{\pm6.1}$ | $42.2_{\pm7.6}$ | $43.6_{\pm4.6}$ | $28.9_{\pm6.0}$ |
| | RR | 7 | $50.5_{\pm3.7}$ | $48.8_{\pm4.8}$ | $50.9_{\pm4.4}$ | $41.9_{\pm3.9}$ | $31.5_{\pm3.3}$ |
| | 3NB | 2 | $84.9_{\pm0.1}$ | $84.9_{\pm0.3}$ | $85.0_{\pm0.5}$ | $84.1_{\pm0.8}$ | $73.9_{\pm1.5}$ |
| | 3NB | 4.5 | $85.9_{\pm0.4}$ | $85.9_{\pm0.4}$ | $85.6_{\pm0.3}$ | $85.6_{\pm0.2}$ | $79.5_{\pm1.6}$ |
| | 3NB | 7 | $86.2_{\pm0.2}$ | $86.1_{\pm0.2}$ | $85.9_{\pm0.3}$ | $85.9_{\pm0.3}$ | $79.9_{\pm1.4}$ |

Table 4: Effect of $\varepsilon_{lbl}$ on accuracy, SST2.

| Kernel | Bitsum | $\varepsilon$ | % Classification accuracy ($\pm$std) with $\varepsilon_{lbl}$: | | | | |
| | | | $\infty$ | 7 | 5 | 3 | 1 |
|---|---|---|---|---|---|---|---|
| Gaussian | RR | 2 | $68.2_{\pm2.9}$ | $66.7_{\pm3.3}$ | $69.0_{\pm2.8}$ | $69.1_{\pm1.7}$ | $61.2_{\pm3.7}$ |
| | RR | 4.5 | $70.0_{\pm4.0}$ | $72.8_{\pm1.8}$ | $70.3_{\pm2.6}$ | $71.4_{\pm2.7}$ | $57.9_{\pm4.9}$ |
| | RR | 7 | $73.1_{\pm1.3}$ | $74.3_{\pm2.5}$ | $72.9_{\pm3.2}$ | $72.1_{\pm1.2}$ | $61.9_{\pm1.8}$ |
| | 3NB | 2 | $72.6_{\pm1.1}$ | $71.8_{\pm4.1}$ | $71.1_{\pm2.7}$ | $70.8_{\pm2.9}$ | $64.6_{\pm2.4}$ |
| | 3NB | 4.5 | $71.5_{\pm3.7}$ | $74.1_{\pm2.0}$ | $72.2_{\pm1.0}$ | $71.5_{\pm2.2}$ | $61.1_{\pm4.6}$ |
| | 3NB | 7 | $70.0_{\pm4.8}$ | $70.9_{\pm1.7}$ | $72.7_{\pm2.8}$ | $71.1_{\pm3.3}$ | $63.6_{\pm2.4}$ |
| IP | RR | 2 | $27.0_{\pm0.8}$ | $27.3_{\pm4.9}$ | $23.0_{\pm1.8}$ | $15.4_{\pm1.8}$ | $7.8_{\pm1.2}$ |
| | RR | 4.5 | $50.5_{\pm3.5}$ | $51.3_{\pm1.6}$ | $46.2_{\pm3.1}$ | $30.0_{\pm2.8}$ | $10.7_{\pm2.0}$ |
| | RR | 7 | $65.4_{\pm2.1}$ | $66.7_{\pm2.0}$ | $63.0_{\pm1.6}$ | $45.2_{\pm3.3}$ | $12.0_{\pm1.8}$ |
| | 3NB | 2 | $92.7_{\pm0.1}$ | $92.8_{\pm0.2}$ | $92.5_{\pm0.2}$ | $91.0_{\pm0.4}$ | $58.9_{\pm1.4}$ |
| | 3NB | 4.5 | $93.1_{\pm0.1}$ | $93.1_{\pm0.0}$ | $93.1_{\pm0.1}$ | $92.5_{\pm0.3}$ | $72.2_{\pm1.2}$ |
| | 3NB | 7 | $93.1_{\pm0.2}$ | $93.1_{\pm0.2}$ | $93.1_{\pm0.1}$ | $92.6_{\pm0.2}$ | $72.9_{\pm1.7}$ |

Table 5: Effect of $\varepsilon_{lbl}$ on accuracy, CIFAR-10

| Kernel | Bitsum | $\varepsilon$ | % Classification accuracy ($\pm$std) with $\varepsilon_{lbl}$: | | | | |
| | | | $\infty$ | 7 | 5 | 3 | 1 |
|---|---|---|---|---|---|---|---|
| Gaussian | RR | 1.5 | $24.1_{\pm1.8}$ | $21.6_{\pm1.6}$ | $21.4_{\pm1.6}$ | $16.9_{\pm2.4}$ | $11.6_{\pm0.6}$ |
| | RR | 3 | $34.8_{\pm3.7}$ | $34.0_{\pm2.4}$ | $36.1_{\pm2.2}$ | $27.5_{\pm0.9}$ | $11.3_{\pm1.2}$ |
| | RR | 4.7 | $46.7_{\pm0.2}$ | $45.2_{\pm2.1}$ | $44.3_{\pm4.0}$ | $36.8_{\pm2.0}$ | $16.7_{\pm1.3}$ |
| | 3NB | 1.5 | $73.4_{\pm1.0}$ | $71.3_{\pm2.0}$ | $72.9_{\pm1.6}$ | $67.8_{\pm1.3}$ | $39.3_{\pm5.4}$ |
| | 3NB | 3 | $73.1_{\pm0.9}$ | $72.2_{\pm1.5}$ | $71.9_{\pm2.2}$ | $67.7_{\pm1.2}$ | $42.8_{\pm3.5}$ |
| | 3NB | 4.7 | $71.5_{\pm1.7}$ | $73.3_{\pm1.6}$ | $72.8_{\pm1.6}$ | $70.4_{\pm1.3}$ | $40.8_{\pm4.1}$ |
| IP | RR | 1.5 | $11.4_{\pm1.6}$ | $10.7_{\pm1.4}$ | $10.6_{\pm2.1}$ | $12.0_{\pm1.6}$ | $10.3_{\pm2.0}$ |
| | RR | 3 | $11.1_{\pm2.6}$ | $9.1_{\pm2.0}$ | $10.6_{\pm1.9}$ | $9.3_{\pm1.3}$ | $10.0_{\pm1.8}$ |
| | RR | 4.7 | $12.0_{\pm1.7}$ | $8.6_{\pm1.8}$ | $10.7_{\pm1.8}$ | $8.7_{\pm1.8}$ | $9.8_{\pm1.5}$ |
| | 3NB | 1.5 | $18.8_{\pm3.3}$ | $19.6_{\pm0.9}$ | $17.9_{\pm3.8}$ | $13.8_{\pm4.2}$ | $12.3_{\pm3.0}$ |
| | 3NB | 3 | $30.4_{\pm4.0}$ | $28.7_{\pm2.4}$ | $26.6_{\pm5.1}$ | $24.4_{\pm2.2}$ | $12.2_{\pm2.1}$ |
| | 3NB | 4.7 | $36.8_{\pm7.8}$ | $37.1_{\pm5.9}$ | $37.8_{\pm2.8}$ | $25.3_{\pm3.6}$ | $13.2_{\pm3.8}$ |

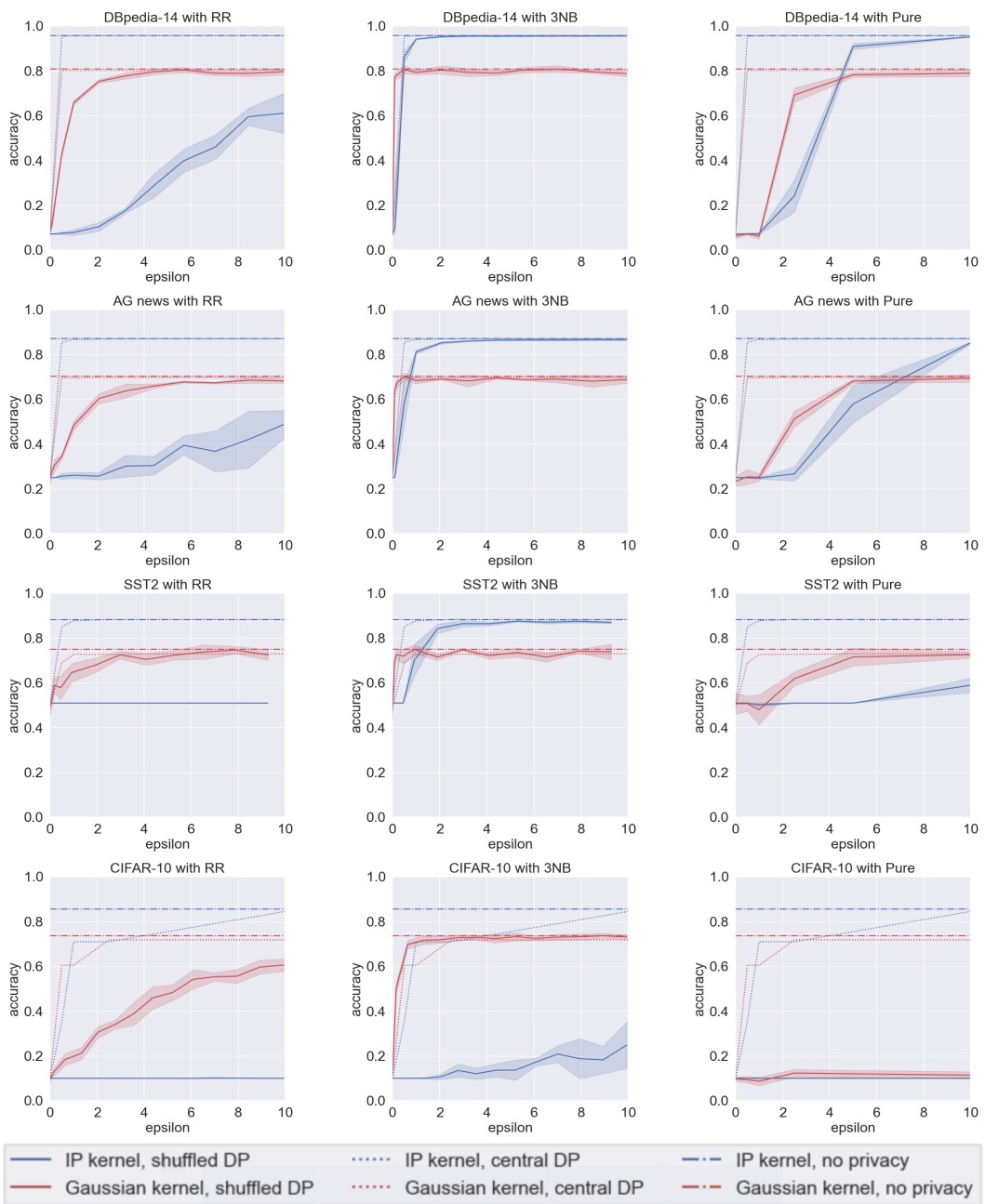

Figure 5: Classification results with $\varepsilon_{\mathrm{lbl}} = 10$

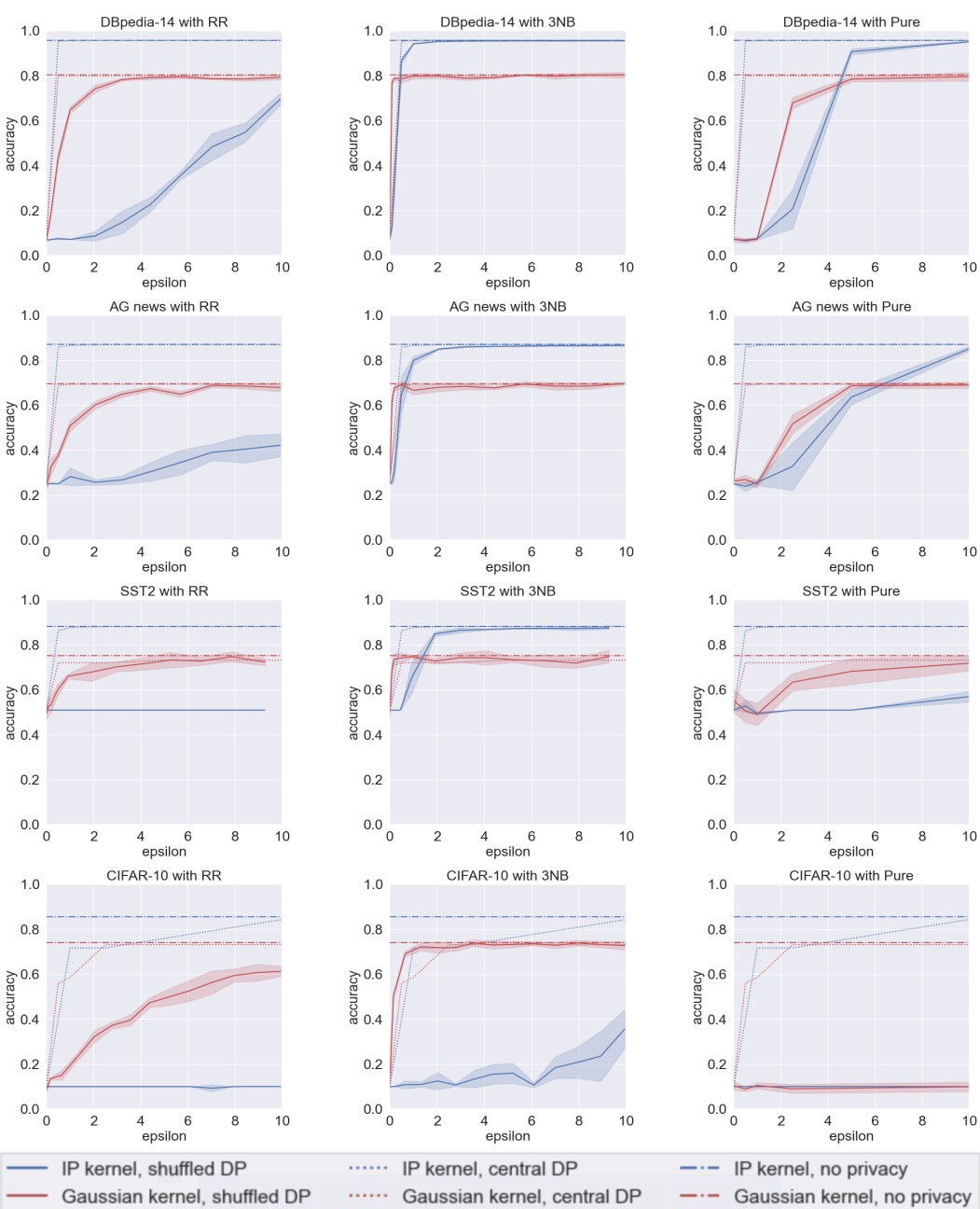

Figure 6: Classification results with $\varepsilon_{\mathrm{lbl}} = 7$

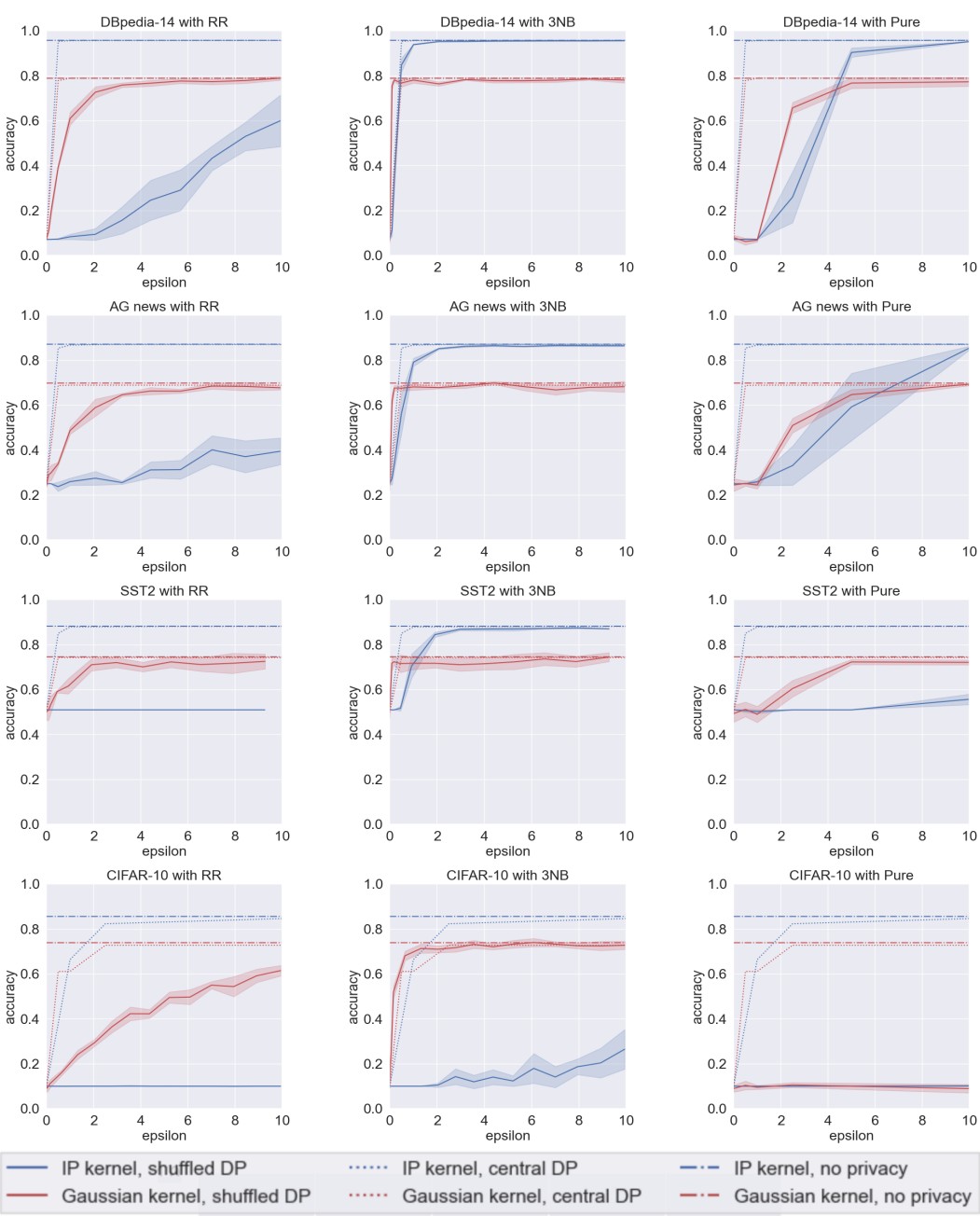

Figure 7: Classification results with $\varepsilon_{\mathrm{lbl}} = 5$ (this is a copy of Figure 1 for convenience)

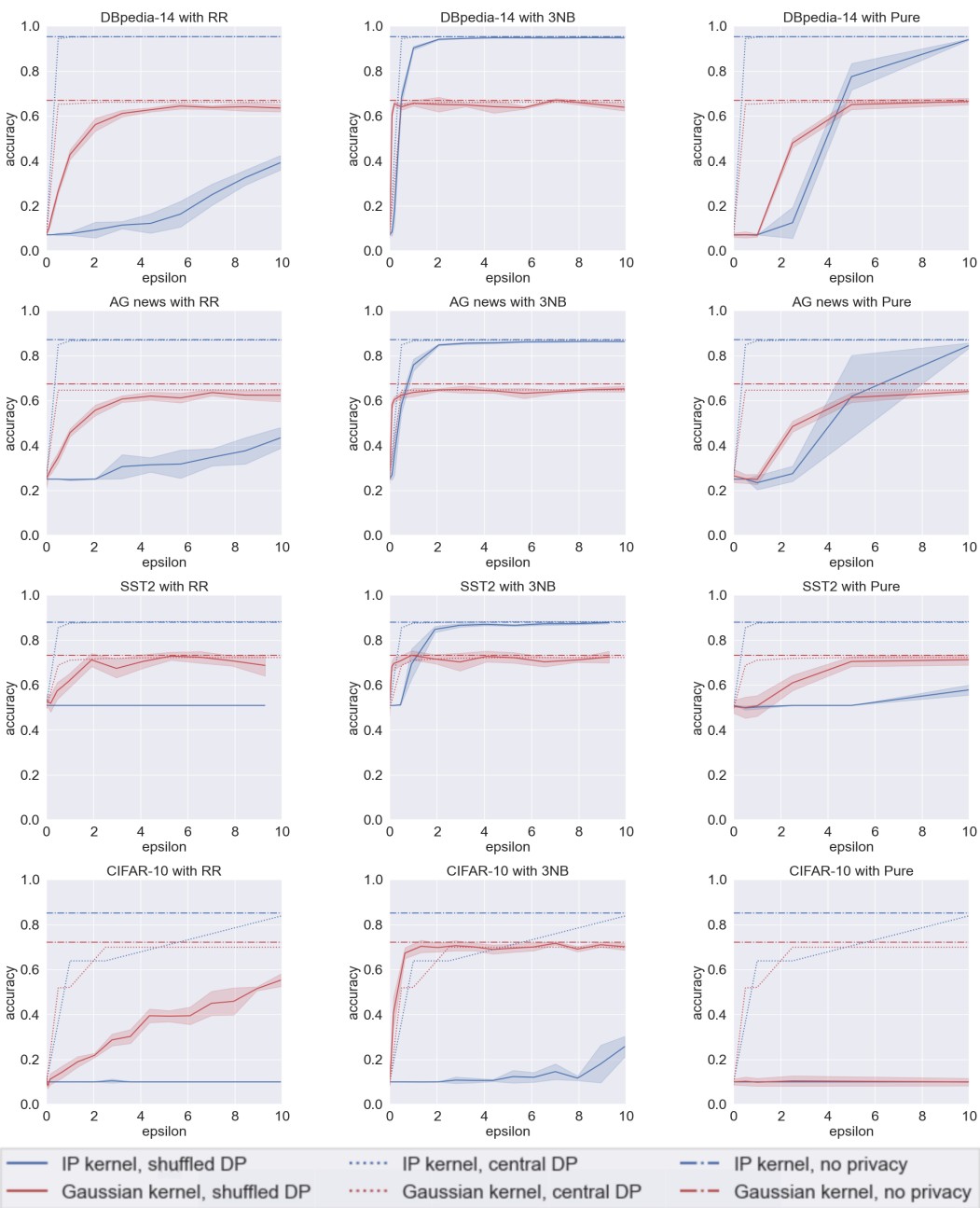

Figure 8: Classification results with $\varepsilon_{\mathrm{lbl}} = 3$

Table 6: Private class decoding results with $\varepsilon_{\mathrm{lbl}} = 5$ and $\varepsilon \approx 5.7$

| Dataset | Class | Bitsum | Gaussian KDE class decoding | IP KDE class decoding |
|---|---|---|---|---|
| DBPedia-14 | Company | RR | seacorp, gencorp, southcorp | xuande, xinmi, xue |
| | | 3NB | europacorp, telekomunikasi, southcorp | companys, railcorp, interactivecorp |
| | | Pure | storebrand, railcorp, onbancorp | companys, companies, nokiacorp |
| | Artist | RR | mandolinist, bosacki, cofounder | ricketson, attributor, vijayendra |
| | | 3NB | author, novelist, writer | author, musician, biographically |
| | | Pure | raymonde, bertogliati, bonifassi | author, roberthoudin, musician |
| | Office holder | RR | legislator, politician, mulroney | cabinetmaker, chatham, provost |
| | | 3NB | ministerpresident, legislator, pastpresident | ministerpresident, legislator, congressperson |
| | | Pure | legislator, reelections, senatorial | ministerpresident, governorsgeneral, legislator |
| | Building | RR | proctorville, connellsville, fargomoorhead | friedrichwilhelmsuniversitt, nordwestmecklenburg, brandenburgbayreuth |
| | | 3NB | galehouse, hyannisport, beaconhouse | churchville, reisterstown, jeffersontown |
| | | Pure | headquarters, northcote, northvale | holyroodhouse, reisterstown, beaconhouse |
| | Village | RR | szewczenko, przodkowo, wodiczko | manasse, jeram, esfahan |
| | | 3NB | khuzistan, wojciechowice, szczawnica | khairabad, kyrghyzstan, kishanganj |
| | | Pure | kleveland, kurdamir, diyarbakirspor | kyrghyzstan, khairabad, yusefabad |
| | Plant | RR | araucariaceae, rubiaceae, araceae | succulents, cunoniaceae, chaetophoraceae |
| | | 3NB | celastraceae, rubiaceae, cactaceae | chenopodiaceae, chaetophoraceae, araucariaceae |
| | | Pure | sapindaceae, violaceae, chenopodiaceae | chenopodiaceae, loranthaceae, gesneriaceae |
| | Film | RR | screenplay, movie, filmore | dishonoring, dishonor, inglorious |
| | | 3NB | toyland, musketeers, imdb | movie, screenplay, biopic |
| | | Pure | screenplay, casablanca, filmmakers | movie, sicario, screenplay |
| | Educational institution | RR | schoolcollege, secondaryschool, boardingschool | humboldtuniversitt, everetts, aleksandrw |
| | | 3NB | bryancollege, boardingschool, schoolship | schoolcollege, boardingschool, polytechnic |
| | | Pure | boardingschool, publicschool, allschool | schoolcollege, boardingschool, polytechnic |
| | Athlete | RR | ajanovic, jovanovski, miloevi | bohuslav, denverbased, petersen |
| | | 3NB | sportsperson, gillenwater, khairuddin | sportsperson, footballer, handballer |
| | | Pure | alifirenko, kovalenko, ilyushenko | laliashvili, jamalullail, footballer |
| | Mean of transport | RR | warship, troopships, aircrafts | curtisswright, veteran, pilotless |
| | | 3NB | fleetness, warship, shipmasters | warship, frigate, torpedoboat |
| | | Pure | warship, battleships, sailed | warship, landcruiser, torpedoboat |
| | Natural place | RR | beringen, freshwater, merideth | bernardini, intermountain, varangians |
| | | 3NB | villeurbanne, riverina, curwensville | river, danube, rivermaya |
| | | Pure | fergushill, danube, waldenburg | floodplain, river, rivermaya |
| | Animal | RR | coraciidae, caeciliidae, cicadellidae | columbellidae, marginellidae, caractacus |
| | | 3NB | carangidae, fasciolariidae, scolopacidae | leiothrichidae, marginellidae, phasianellidae |
| | | Pure | coccinellidae, coraciidae, cardinalidae | dendrobatidae, margaritidae, catostomidae |
| | Album | RR | discography, vocals, stereophonics | album, korn, groupie |
| | | 3NB | album, instrumentals, tracklist | album, discography, tracklist |
| | | Pure | discography, pledgemusic, vocals | album, discography, allmusic |
| | Written work | RR | biographies, storybook, author | booksurge, huilai, huizhou |
| | | 3NB | bibliography, biographies, autobiography | nonfiction, author, biographies |
| | | Pure | wittgenstein, werman, fangoria | nonfiction, author, bibliography |
| AG news | Sports | RR | winningest, standings, playoff | mccolm, mccartt, inconclusive |
| | | 3NB | huels, kurkjian, darrington | playoff, championship, standings |
| | | Pure | gallardo, unfit, basketball | playoff, postseason, runsgriffey |
| | Business | RR | theba, buybacks, kulikowski | surcharging, nonhazardous, surcharges |
| | | 3NB | retrials, clawbacks, revaluation | nasdaq, enron, divestitures |
| | | Pure | nasdaq, exxonmobil, exxon | nasdaq, divestitures, nyse |
| | World | RR | terrorist, bombings, ilghazi | ppas, atpranking, deng |
| | | 3NB | hostages, baghdadi, nabaa | terrorists, militants, qaeda |
| | | Pure | qaeda, iraqstld, antifur | qaeda, ceasefire, intifadas |
| | Sci/Tech | RR | ibm, loango, launchpads | ough, iebc, oul |
| | | 3NB | microsoft, viacom, protv | microsoft, ibm, lucenttech |
| | | Pure | edgeware, proximity, shareware | ibm, infotrends, suntec |
| SST2 | Negative | RR | overstating, dreadful, awfulness | gameplay, tactics, underplaying |
| | | 3NB | derailments, vagueness, breakage | blandness, dramaturgy, comedy |
| | | Pure | perversities, sentimentalism, overthinking | tragedy, blandness, melodrama |
| | Positive | RR | fervor, phenomenom, pageantry | embeddable, imbedding, embed |
| | | 3NB | cinema, screenplays, films | evocative, salaciousness, theatricality |
| | | Pure | majestically, dramatization, shrewdness | memorability, evocative, masterpieces |

Table 7: Private class decoding results with $\varepsilon_{\text{lbl}} = 5$ and $\varepsilon \approx 4.4$

| Dataset | Class | Bitsum | Gaussian KDE class decoding | IP KDE class decoding |
|---|---|---|---|---|
| DBPedia-14 | Company | RR | companys, manufacturera, comcorp | airgroup, aerosystems, bluepoint |
| | | 3NB | manufactories, subsidiaries, originators | companys, railcorp, baycorp |
| | | Pure | railcorp, companys, comapny | companys, companies, manufactories |
| | Artist | RR | singer, musician, ewan | marxer, auditioner, surinder |
| | | 3NB | balladeer, musician, artiste | author, musician, novelist |
| | | Pure | author, artist, composers | author, biographically, musician |
| | Office holder | RR | mcclelland, mclellan, dreiberg | janetta, janette, jadakiss |
| | | 3NB | representant, representan, representantes | legislator, ministerpresident, congressperson |
| | | Pure | bashiruddin, ministerpresident, lazarescu | politician, ministerpresident, liberhan |
| | Building | RR | woodville, marksville, douglassville | poteet, hocutt, chestnutt |
| | | 3NB | stationhouse, randallstown, dovercourt | churchville, chapeltown, beaconhouse |
| | | Pure | headquarter, hyattsville, weaverville | holyroodhouse, fenchurch, charleswood |
| | Village | RR | krakowiak, krzynowoga, lubliniec | kieslowski, radiolocation, blenkiron |
| | | 3NB | kyrghyzstan, khazakstan, diyarbakir | kyrghyzstan, khairabad, khuzistan |
| | | Pure | taleyarkhan, voivodeship, mieszkowice | khuzestan, kyrghyzstan, diyarbakir |
| | Plant | RR | saxifragaceae, loranthaceae, sapotaceae | lauraceae, loganiaceae, annonaceae |
| | | 3NB | orobanchaceae, mycenaceae, bromeliaceae | chenopodiaceae, araucariaceae, loranthaceae |
| | | Pure | chenopodiaceae, podocarpaceae, cactaceae | cupressaceae, rubiaceae, chaetophoraceae |
| | Film | RR | biopic, imdb, screenplay | dollywood, isoroku, rakotomanana |
| | | 3NB | biopic, movie, silmarillion | movie, screenplay, biopic |
| | | Pure | biopic, cinemax, films | movie, sicario, biopic |
| | Educational institution | RR | ucda, madrassa, polytechnic | write, reflectometry, chathams |
| | | 3NB | schoolcollege, polytechnic, fachhochschule | schoolcollege, boardingschool, polytechnic |
| | | Pure | eduniversal, universits, universitat | schoolcollege, boardingschool, publicschool |
| | Athlete | RR | borgne, romanowski, brzezinski | sobolewski, khatemi, wlosowicz |
| | | 3NB | laliashvili, gianluigi, pirlo | sportsperson, handballer, ivanovic |
| | | Pure | pejaevi, milanovic, tomashova | konashenkov, laliashvili, kalynychenko |
| | Mean of transport | RR | battleships, navymarine, landcruiser | pinezhsky, pisetsky, ilyinsky |
| | | 3NB | spitfires, troopships, maersk | warship, landcruiser, frigate |
| | | Pure | warship, frigate, torpedo | warship, frigate, landcruiser |
| | Natural place | RR | krauchanka, gaucelm, kotonowaka | halethorpe, mapplethorpe, chloropaschia |
| | | 3NB | danube, tributary, vilfredo | river, rivermaya, rivervale |
| | | Pure | soligorsk, vassilakis, nordgau | floodplain, danube, river |
| | Animal | RR | coraciidae, glareolidae, phyllostomidae | mollusc, motacillidae, molluscan |
| | | 3NB | paludomidae, acrolepiidae, discodorididae | leiothrichidae, marginellidae, catostomidae |
| | | Pure | marginellidae, riodinidae, orthogoniinae | mantellidae, catostomidae, coraciidae |
| | Album | RR | album, vanilli, europop | melancholy, poetica, majra |
| | | 3NB | allmusic, remixes, remixed | album, discography, allmusic |
| | | Pure | housemusic, tracklist, musicology | album, discography, tracklist |
| | Written work | RR | booknotes, pulitzerprize, apocryphally | terrors, sarkies, dementyeva |
| | | 3NB | novelistic, magazine, novelist | nonfiction, author, novelist |
| | | Pure | authorites, novelette, novelist | nonfiction, bibliography, author |
| AG news | Sports | RR | lose, playoff, sportschannel | eddard, frankenfish, paeonian |
| | | 3NB | cbssportscom, hof, injuries | playoff, semifinalists, championship |
| | | Pure | byrd, garvin, deq | injury, guardino, tiedown |
| | Business | RR | gencorp, walkout, comstock | citywest, epson, arkwright |
| | | 3NB | opec, sirri, toyota | nasdaq, stockholders, divestitures |
| | | Pure | nonactors, goldcorp, archconfraternity | outbids, stockpiling, outselling |
| | World | RR | collusion, dahle, kejie | yawner, oswalt, russ |
| | | 3NB | islamiah, taliban, hurghada | hamas, qaeda, taliban |
| | | Pure | bomb, nomination, warmongering | baghdadi, occupiers, militants |
| | Sci/Tech | RR | baidu, microsoft, tencent | multicamera, intercambio, videoconferences |
| | | 3NB | ibm, httpwwwdaimlerchryslercom, computer-ware | microsoft, ibm, lucenttech |
| | | Pure | infotrends, ati, infogear | redesigns, ibm, lucenttech |
| SST2 | Negative | RR | portrayal, dreariness, bleakness | repeated, twicetobeat, ringed |
| | | 3NB | murkiness, unexciting, vapidity | comedy, dramaturgy, dramaturgical |
| | | Pure | dramaturgy, portrayals, dramatising | unpleasantries, unpleasantness, deadness |
| | Positive | RR | reworded, comedies, critiques | rastignac, ruderman, gritschuk |
| | | 3NB | memorability, rausing, miserables | masterpieces, salaciousness, evocative |
| | | Pure | intiative, artistical, screenplays | vividness, evocative, presence |

Table 8: Private class decoding results with $\varepsilon_{lbl} = 5$ and $\varepsilon \approx 3.2$ (full version of Table 1)

| Dataset | Class | Bitsum | Gaussian KDE class decoding | IP KDE class decoding |
|---|---|---|---|---|
| DBPedia-14 | Company | RR | vendors, gencorp, servicers | firesign, wnews, usos |
| | | 3NB | molycorp, newscorp, mediacorp | companys, alicorp, interactivecorp |
| | | Pure | ameritech, alicorp, newscorp | alibabacom, oscorp, companies |
| | Artist | RR | author, aristizabal, levesongower | catalani, macki, bacashihua |
| | | 3NB | artist, lyricists, musician | author, musician, roberthoudin |
| | | Pure | author, originator, mikhaylovsky | musician, artist, jacquesfranois |
| | Office holder | RR | louislegrand, legislator, lawmaker | sulphide, exclude, sulked |
| | | 3NB | patiashvili, kumaritashvili, biographers | ministerpresident, legislator, congressperson |
| | | Pure | polinard, bobrzaski, politician | ministerpresident, politician, polian |
| | Building | RR | reisterstown, benenson, hellertown | opened, mastered, poegaslavonia |
| | | 3NB | frenchtown, brookeville, kenansville | beaconhouse, manorville, reisterstown |
| | | Pure | huntingtonwhiteley, wrightstown, randallstown | hyattsville, roxboro, reisterstown |
| | Village | RR | lalganj, balrampur, manikganj | baluchestan, jagiellonia, nidderdale |
| | | 3NB | pazardzhik, tzintzuntzan, khuzistan | kyrghyzstan, kalinske, kalinski |
| | | Pure | poniewozik, mieszkowice, czerniewice | kazemabad, diyarbakir, khoramabad |
| | Plant | RR | chaetophoraceae, gentianaceae, rutaceae | chilensis, surinamensis, tampines |
| | | 3NB | cupressaceae, chaetophoraceae, podocarpaceae | chenopodiaceae, araucariaceae, loranthaceae |
| | | Pure | asclepiadaceae, cupressaceae, gentianaceae | chaetophoraceae, chenopodiaceae, araucariaceae |
| | Film | RR | biopic, movie, screenplay | kaptai, kakhi, kaloi |
| | | 3NB | filmography, vanya, ghostbusters | movie, filmography, screenplay |
| | | Pure | filme, movie, videodrome | movie, film, filmmakers |
| | Educational institution | RR | schoolcollege, boardingschool, allschool | eastern, marykane, kbe |
| | | 3NB | boardingschool, schoolcollege, publicschool | schoolcollege, boardingschool, polytechnic |
| | | Pure | schoolcollege, polytechnic, qschool | schoolcollege, publicschool, boardingschool |
| | Athlete | RR | kovaleski, kaessmann, miroshnichenko | torstensson, torstenson, torlakson |
| | | 3NB | fabianski, tarnowski, bochenski | sportsperson, laliashvili, konashenkov |
| | | Pure | rightfielder, leftfielder, konashenkov | lukasiewicz, sportsperson, marcinkiewicz |
| | Mean of transport | RR | warship, frigate, steamships | latrodectus, laax, herx |
| | | 3NB | landcruiser, warship, landships | warship, frigate, landcruiser |
| | | Pure | battlecruiser, warship, hmso | warship, landcruiser, connaught |
| | Natural place | RR | tributary, riverina, river | bimota, miercoles, mientras |
| | | 3NB | langenlonsheim, nordwestmecklenburg, schweinfurt | rivermaya, river, danube |
| | | Pure | riverbeds, azkoitia, zaporozhian | lakernotes, river, riverina |
| | Animal | RR | carangidae, caeciliidae, arctiidae | taricani, tardio, kambona |
| | | 3NB | leiothrichidae, coleoptera, acrolepiidae | marginellidae, limoniidae, catostomidae |
| | | Pure | coraciidae, poeciliidae, acrolepiidae | caeciliidae, heliozelidae, lasiocampidae |
| | Album | RR | discography, sevenfold, album | roadster, approximant, pantocrator |
| | | 3NB | discography, album, allmusic | album, discography, allmusic |
| | | Pure | discography, album, allmusic | album, decemberists, song |
| | Written work | RR | nonfiction, encyclopedia, nonfictional | sociolinguist, becc, sociolegal |
| | | 3NB | author, magazine, novelist | nonfiction, author, biographies |
| | | Pure | reganbooks, novelettes, fourbook | synopsis, nonfiction, biographies |
| AG news | Sports | RR | vizner, runnerups, dietrichson | ongeri, grandi, zarate |
| | | 3NB | injury, semifinalists, finalists | semifinalists, championship, standings |
| | | Pure | pensford, rematches, undefeated | chauci, teammates, nith |
| | Business | RR | repurchases, downtrend, equitywatch | sneed, timesnews, anxiousness |
| | | 3NB | enrononline, investcorp, comcorp | stockholders, nasdaq, marketwatchcom |
| | | Pure | corporations, consolidations, consolidated | merger, divestiture, stockholders |
| | World | RR | iraqi, hamas, darfur | tym, asg, tyo |
| | | 3NB | hezbollah, hamas, iraqstld | hamas, terrorists, baghdadi |
| | | Pure | kutayev, qaeda, yanukovych | shamkir, barricading, samaritans |
| | Sci/Tech | RR | snopes, cyberworks, hacktivists | meanings, collegefootballnewscom, multipolarity |
| | | 3NB | ibm, thermedics, flextech | microsoft, ibm, accenture |
| | | Pure | feedbacks, companywide, eurogroup | movedtech, techcrunch, swindlers |
| SST2 | Negative | RR | beguile, inception, shallow | manipulating, uncouple, dissects |
| | | 3NB | melodrama, rawness, blandness | comedy, tastelessness, uneasiness |
| | | Pure | chumminess, meaningfulness, mootness | absurdities, chastisement, absurdity |
| | Positive | RR | kindliness, pleasantness, entertaining | enjoyments, academie, amusements |
| | | 3NB | salacious, movie, majestic | salaciousness, theatricality, memorability |
| | | Pure | spiritedness, spirited, perspicacious | exorcisms, fairytales, revisiting |

