# OpenReview forum: "Learning from End User Data with Shuffled Differential Privacy over Kernel Densities"
_ICLR.cc/2025/Conference — ICLR 2025 Spotlight_

### Official Review · Reviewer_Nw6K · 2024-11-02

**Soundness:** 3
**Presentation:** 3
**Contribution:** 2
**Rating:** 8
**Confidence:** 2

**Summary:**

This paper studies private kernel density estimation and the privacy at the user end is considered. The methodology is with the protocol of the shuffled-DP: each user first randomize her own data and then a shuffler will collect the randomized messages from all users and shuffle them in a random order. The basic module is the bitsum in the shuffled DP where each user has one bit and the set of bit after shuffling keeps well the sum of all bits. The main algorithm is designed to call this module towards the private kernel estimation. The paper provides the theoretical guarantee for the algorithm. In the experiment, the proposed algorithm under shuffled-DP is compared with the algorithm at central-DP in two downstream tasks classification and class decoding.

**Strengths:**

1. The flow of the writing and the clarity are great. The background such as the framework of shuffled DP is mostly well-introduced; I do have questions for some concepts (see the question section), but they are not influencing my overall understanding for the method.
2. The analysis of proposed algorithm seems solid as presented in theorem 3.2 and theorem 3.3.
3. The empirical evaluation is systematic. It is located to two downstream tasks of the private kernel estimation.

**Weaknesses:**

As this paper is the first paper (to my best knowledge) to study private kernel estimation with the shuffled DP protocol, I think more comparison with other settings can enhance the understanding what role shuffled DP plays. In details:
1. How is the proposed method compared with central DP in terms of theoretical guarantees? Would this be aligned with the empirical comparison as shown in the experiment section?
2. Another popular framework is to leverage secure multiparty computation (MPC) to compute sums of values from each user [1]. What is the advantage and limitations between these two framework in terms of utility and communication cost?

[1] Bonawitz, Keith, et al. "Practical secure aggregation for privacy-preserving machine learning." proceedings of the 2017 ACM SIGSAC Conference on Computer and Communications Security. 2017.

**Questions:**

Please see the weaknesses section. Moreover, I have some questions of some mentioned concepts:
1. Bit-width is introduced in line 239-240. What is the exact definition of it? In the theorem, "the protocol has optimal bit-width 1" is stated; what does this mean?
2. $S$ is introduced in Definition 2 as a property to describe the kernel. It does not appear in the algorithm's description but seems to be an important factor for the theoretical guarantees, which occurs in both DP's guarantee and the utility upper bound. Could you explain what role $S$ plays to bring these effects on the theoretical results?

---

> ### Author Response · Authors · 2024-11-18
>
> We thank you for the review.
>
> > How is the proposed method compared with central DP in terms of theoretical guarantees? Would this be aligned with the empirical comparison as shown in the experiment section?
>
> Under $(\varepsilon,\delta)$-DP, the central DP KDE error is $O(\sqrt{\log(1/\delta)}/(\varepsilon n))$. In our shuffled DP KDE results, 3NB has the same error asymptotically (this is the statement of Theorem 3.3), while RR has error larger by another factor of $\sqrt{\log(1/\delta)}$ (with the gain of smaller communication cost than 3NB). Under pure $\varepsilon$-DP, the central DP KDE error is $O(1/\sqrt{\varepsilon n})$, and our shuffled DP result with Pure has the same error asymptotically. These results align with the experimental section, where all the tested DP methods (central and shuffled) converge to the accuracy of exact DP once $\varepsilon$ and $n$ are large enough.
>
>
> > Another popular framework is to leverage secure multiparty computation (MPC)
>
> DP and MPC handle different goals in data protection, and are often used together, especially in shuffled DP – where the shuffler, which is considered “trusted” in the shuffled DP model, is implemented by MPC to achieve this trustworthiness. For example, the paper [1] you mention is referenced in Kairouz et al., 2021a;b as a possible implementation of the shuffler.
>
> In general, MPC protects the computation *process*, while DP protects the computation’s *output*. Take bitsum as an example: each of $n$ users in a communication network holds a private bit, and they want to compute and release the sum, without leaking information about their individual bits.
> * They could send their bits to the analyzer, who would compute the sum and add Laplace noise to ensure DP. The output is DP, but an adversary who eavesdropped on the communication has learned all the individual bits anyway. This is DP without MPC.
> * They could use MPC to compute and release the exact sum. MPC ensures that the eavesdropping adversary has learned nothing from the communication except for the intended output, i.e., the sum. But the sum itself is not private, since as a deterministic function of the inputs, it may leak information about individual bits (for example, the adversary now knows whether all users had zero bits or not, which violates DP). This is MPC without DP.
> * They could first locally add noise to their bits (say, flip them with some probability, as in RR), then use MPC to compute and release the sum of the noisy bits. This is essentially the shuffled DP model. If the local noise is chosen such that the sum of noisy bits is DP — which is what shuffled DP bitsum protocols are formally designed to achieve — then the adversary has learned nothing (up to the $\varepsilon$ privacy budget of DP) from neither the communication nor the output. This is the combination of DP and MPC.
>
> Thus, the shuffled DP model can be understood as an abstraction of the MPC portion of the protocol into a “trusted shuffler”, or alternatively, as an abstraction of an MPC protocol with a DP guarantee on its output. MPC in this discussion can be replaced with other notions of secure computation (which protect the “process”).
>
> > Bit-width is introduced in line 239-240. What is the exact definition of it?
>
> For a shuffled DP protocol $\Pi$, the bitwidth $\ell=\ell(\Pi)$ is defined as the number of bits in the binary representation of each of its numerical values. We distinguish here between numerical values, which are ones used for arithmetic computations, and non-numerical values (like user IDs or enumerative indices, which are often written as integers for convenience, but need not be involved in arithmetics). It is possible that $\ell=\infty$ (say, if $\Pi$ uses real numbers in messages).
>
> The reason this parameter was introduced and bounded in prior work on shuffled DP is that in practice, the shuffler is often implemented with MPC using modular arithmetics (the paper [1] you cited is an example). To enable this, $\ell$ must be finite. Furthermore, since the arithmetics in MPC is modulo $2^\ell$, the practical running time scales exponentially with the bit-width, and therefore it is important to control. Our DP KDE protocols have bit-width 1, as all the numerical values they communicate in messages are integers of magnitude 1.

---

> > ### Author Response · Authors · 2024-11-18
> > **Continued response**
> >
> > (Continued answer)
> >
> > > Could you explain what role $S$ plays to bring these effects on the theoretical results?
> >
> > $S$ is the sparsity parameter LSQ, which guarantees that for every $f,g$ and $x,y$, $f(x)$ and $g(y)$ are $Q$-dimensional vectors with at most $S$ non-zero coordinates.
> >
> > Role in privacy: Each user with input $x$ will share her $f(x)$ with privacy. She will do this by participating in $Q$ shuffled DP bitsum protocols, one per coordinate of $f(x)$, so naively her privacy loss might scale with $Q$. However, since she actually contributes information to only $S$ of these protocols (her non-zero entries), intuitively her privacy loss should only scale with $S$. This is rigorously true, and this is why in Theorem 3.2 we have $\varepsilon\approx\varepsilon_0S$ rather than $\varepsilon$ depending on $Q$.
> >
> > Role in accuracy: The analyzer will privately learn some $Q$-dimensional vector $\widetilde F$ from the users (one coordinate per bitsum protocol), and will return $\widetilde F^Tg(y)$ as the KDE estimate for a query $y$. Due to bitsum protocol errors, each coordinate of $\widetilde F$ has some noise. So when we sum those $Q$ coordinates in the inner product $\widetilde F^Tg(y)$, naively the total noise might scale with $Q$. However, since $g(y)$ has only $S$ non-zero entries, we actually only sum $S$ of these noisy coordinates, so the total noise should only scale with $S$. This is why in Theorem 3.2 the supRMSE scales with $S$ and not with $Q$.
> >
> > The above arguments hold without the users or the analyzer doing anything to achieve them – they hold “automatically” whenever sparsity is satisfied by the LSQ family, and thus you are right that $S$ does not show up in the algorithm. It plays a role in the analysis (proof of Theorem 3.2) that bounds the privacy loss and the total error, where the above arguments are formalized.

---

> > > ### Comment · Reviewer_Nw6K · 2024-11-21
> > >
> > > Thank authors for their responses, which addressed my questions. I have raised my score accordingly. I also lowered my confidence, because I have less expertise to evaluate the novelty in the technical part related to MPC.

---

### Official Review · Reviewer_3tVM · 2024-11-04

**Soundness:** 3
**Presentation:** 3
**Contribution:** 3
**Rating:** 8
**Confidence:** 4

**Summary:**

The paper focuses on learning a kernel density estimator (KDE) under shuffled differential privacy (SDP), while the learned KDEs are used for a downstream classification task. The proposed method first estimates noisy class counts under local DP, and then learns KDE for each class using the noisy counts (and the corresponding class assignments) for calibrating SDP noise level. The method for estimating KDE is based on the notion of locality-sensitive quantization (LSQ), which also allows the authors to prove general as well as more specific utility bounds (e.g. for Gaussian kernel with random Fourier features). The authors experimentally demonstrate their proposed method on several data sets using various existing SDP summation protocols together with Gaussian and inner product kernels.

**Strengths:**

i) The proposed method seems to be interesting and novel in the SDP setting.

ii) The paper is mostly well-written and nice to read, with no obvious major oversights.

iii) The code is included with the submission (at least partly; e.g., as there is no requirements.txt or equivalent, getting the code to run would take some amount of effort).

**Weaknesses:**

i) While the stated goal is to do learning in a "single-shot" manner, i.e., without clients participating (iteratively) in training beyond just choosing to send their data, the proposed approach still involves 2 communication rounds (getting noisy counts for the classes, then running the KDE protocol with noisy class sizes) and a non-trivial amount of compute spend on client-side (run some pretrained model as feature-extractor, run Alg.1 with given hypers).

ii) See comments below for some additional specific concerns.

**Questions:**

## Update after the discussion:

The authors have mostly responded well to all my expressed concerns; I still have some minor disagreements, e.g., over the class decoding, but these are minor points and should not prevent accepting the paper. I have increased my score accordingly.

## Original questions:

In decreasing order of importance:

1) On the distinction between classification and class decoding, e.g. lines 315-27, 526-29: I am not sure how meaningful this difference actually is; I would expect that to be able to classify examples well, the KDE $\tilde K_c$ for any given class $c \in [m]$  should give higher scores than any other KDE for that class. This seems pretty much the same things as any given KDE $\tilde K_c$ being able to give high scores for inputs from the matching class $c$ and lower scores to inputs from any other class. Whether or not this matches what humans would consider to be semantically relevant for the classes seems like quite a fuzzy claim.

2) Experimental details: please specify how all hyperparameters have been set, how many seeds you used etc. Also, please add some error metrics (sem, std or similar) to the results.

3) How much utility do the noisy counts via LDP cost? Please add an ablation study (at least for some experimental setting) using the actual true counts and comparing to the proposed full method with LDP counts (using same noise levels for the shuffle protocol).

4) Thms 3.2, 3.3: please clarify in what sense the bit-width 1 is optimal (e.g. communication, utility, both)?

5) Lines 312-14: there are some proposed DP-kNN methods, so this claim does not seem to be true (although I haven not read the said papers in detail).

6) Please add an additional step after line 913 to be clearer on the proof.

## Minor typos, comments etc.

I do not expect any acknowledgement or comment on these, just fix when appropriate:
* typos: lines 109-110, 775-76, 1063-64, 1139-40, 1160-61
* Why suppress eq numbers? Would be more straightforward to ref these using numbers instead of lines.
* To make the result figures more readily interpretable, it would be good to add some more basic info to the captions (e.g. $\delta$, how many repeats).

**Details Of Ethics Concerns:**

I have not specific ethical concerns for this paper.

---

> ### Author Response · Authors · 2024-11-18
>
> We thank you for the review.
>
> > On the distinction between classification and class decoding
>
> Perhaps the clearest way to see the distinction, is to consider class decoding without classification at all. Suppose the users have unlabeled private texts, with some prevalent topical theme unknown to the learner, say, films. Not all texts need to be about films, but suppose most are. The analyzer learns a DP KDE representation $\widetilde K$ of the embedded texts. Note that there is no classification component, since there is only “one class”. The analyzer never sees labeled negative examples, and is never told about any text whether it belongs to the topical theme or not. The setting is completely unsupervised.
>
> Still, by class decoding, the analyzer will give the highest scores to dictionary terms like “biopic, movie, screenplay”, and can discern the topical theme of the user texts from the private representation $\tilde K$.
>
> This is generally expected if $\tilde K$ was a non-private KDE estimate, and also if $\tilde K$ was learned under central DP, since in both those cases the learner constructs $\tilde K$ from plain unprotected texts (or their embeddings), and she would naturally incorporate their prevalent themes into the $\widetilde K$. Simply put, she learns their content by just reading them. The difference we wish to highlight is that in the shuffled DP setting, the learner achieves this without seeing any unprotected text, in the strong sense that she provably (by DP) cannot glean substantial information about any of the texts. In a sense, she can tell what the texts are generally about without reading any of them. This is, of course, made possible because each privatized text does leak “a little bit” of its content (within the privacy budget), and if sufficiently many of them are topically consistent, the shuffled DP KDE mechanism detects and expresses the accumulated topical signal.
>
> > Experimental details: please specify how all hyperparameters have been set, how many seeds you used etc. Also, please add some error metrics (sem, std or similar) to the results.
>
> The only hyperparameter is the number of repetitions/features $I$ in the protocol (line 221), which we set to $d$ (the embedding dimensions) for the most direct comparison with the non-private KDE baseline (which just uses the $d$ original embedding features). The shaded regions around the curves in the plots show the std over 5 random seeds.
>
> > How much utility do the noisy counts via LDP cost? Please add an ablation study (at least for some experimental setting)
>
> We have added this to the uploaded version, please see Tables 2-5 on pages 25-26.
>
> > in what sense the bit-width 1 is optimal (e.g. communication, utility, both)?
>
> We give the definition of bitwidth in an answer to Nw6K, who raised a similar question. The role of bit-width is to ensure that the secure aggregation part of the protocol (which is abstracted as the shuffler) can be made practically efficient. It may generally degrade communication and utility as follows: The communication clearly grows with the bit-width (though it is not governed by it, as it also grows with the non-numerical values in protocol). As for utility, since large or infinite bit-width may make the protocol infeasible or impossible, it is common to cap the bit-width by truncating the numerical values in the protocol to a lower bit precision. This creates another source of error (rounding errors), degrading utility. The best possible bit-width 1, which our protocol attains, has no degrading impact on neither the communication nor the utility.
>
> > Please add an additional step after line 913 to be clearer on the proof.
>
> We have expanded this in the uploaded version (it is now line 918).

---

> > ### Comment · Reviewer_3tVM · 2024-11-22
> > **Some additional questions**
> >
> > Thanks for the clarifications and effort, as a minor follow-up:
> >
> > i) On the class decoding: so if I understand you correctly, what you basically mean by class decoding is just querying a given density estimator for the higher-density regions of space, e.g., with text/word embedding data meaning higher frequency words/embeddings? I fail to see why centralized DP should be somehow fundamentally different from shuffled DP in this case, as also under CDP, the learner can't simply read the data and learn it due to DP; I think in this case CDP learner can work pretty much the same way as a single local DP mechanism under shuffled DP but with less randomization. However, this seems like a minor issue overall.

---

> > > ### Author Response · Authors · 2024-11-23
> > >
> > > Thank you for following up.
> > >
> > > >  what you basically mean by class decoding is just querying a given density estimator for the higher-density regions of space, e.g., with text/word embedding data meaning higher frequency words/embeddings?
> > >
> > > Yes -- with the small subtlety that the top-ranked terms are not necessarily those with the highest individual frequencies (as in a histogram), but the highest density, since the KDE estimator is learned on entire text embeddings. Say, when we query the word “film” we get the estimated KDE at the embedding vector $v_{\text{film}}$ of “film”, even though the learner hasn’t seen this particular vector in the dataset (which doesn’t contain text records with only one word). The KDE mass at $v_{\text{film}}$ accumulates from embeddings of user texts that may or may not contain the word “film”, but in either case their content was semantically related to “film” so the pretrained embedding model embedded them near $v_{\text{film}}$ in $R^d$.
> > >
> > > > why centralized DP should be somehow fundamentally different from shuffled DP
> > >
> > > The distinction is a bit subtle – it would help to differentiate between three parties in the pipeline:
> > > 1. The users who hold the data (they run the “randomizer” in Alg 1)
> > > 2. The analyzer who computes $\widetilde K$ from the data and releases it (the “analyzer” in Alg 1)
> > > 3. The querier who evaluates the released $\widetilde K$ on the vocabulary (the “KDE query” part in Alg 1)
> > >
> > > In central DP, DP is injected between 2 and 3. In shuffled or local DP, it is injected between 1 and 2. This is why we did not differentiate between the analyzer and the querier so far in the discussion of class decoding in shuffled DP, as both are post-processing anyway; but it is helpful for the comparison with central DP. We just intended to point out that in CDP, the analyzer computes $\widetilde K$ before privacy was imposed, meaning she has full access to the unprotected user data (say, she can compute a very high quality KDE estimator without any privacy considerations, and only then privatize it), whereas in shuffled DP, she computes $\widetilde K$ after privacy was already imposed, and thus her access to the raw data is much more limited (say, she cannot compute the exact KDE). Yet in both cases, meaningful class decoding can still be achieved by of querier. Thus, the models are different from the point of view of the analyzer, even though they are the same from the point of view of the querier.

---

> ### Author Response · Authors · 2024-11-18
> **Continued response**
>
> > there are some proposed DP-kNN methods
>
> We suppose you mean the line of work [A,B,C]. While they use the kNN terminology as motivation, the problem they actually solve is not nearest neighbor search (NNS), but near neighbor **counting** (which they use as a DP stand-in of NNS, not unlike how we use KDE). The near neighbor counting problem (NNC) is the following: given a dataset $X\subset\mathbb R^d$ and a scale parameter $r>0$, build a data structure that given a query $y\in\mathbb R^d$, reports the number of points in $X$ at distance at most $r$ from $y$. We will denote this number by $|B_X(y;r)|$. In DP NNC, which these papers solve, the goal is for the data structure to report private estimates of $|B_X(y;r)|$.
>
> NNC is very close to KDE, in fact (if we normalize the counts by $|X|$) it is a special case of KDE, with the “step kernel” $\mathrm{\mathbf{k}}_r(x,y)=1$ if $\lVert x-y\rVert_2\leq r$ and $\mathrm{\mathbf{k_r}}(x,y)=0$ otherwise. $r$ plays the same role bandwidth plays in kernels.
>
> Let us explain why this line of work is not suitable for our goals. We focus on [C], which is the SOTA for high dimensions. First, in terms of accuracy, they are unable to estimate $|B_X(y;r)|$ privately with additive error. Instead, they introduce another approximation parameter $c>0$, and their central DP mechanism reports a count between $|B_X(y;r)|$ and $|B_X(y;cr)|$, with additive error that grows like $n^{O(1/c^2)}/\varepsilon$. Formally, the normalized count is guaranteed to be between $\tfrac1n|B_X(y;r)| - 1/\varepsilon n^{1-O(1/c^2)}$ and  $\tfrac1n|B_X(y;cr)| + 1/\varepsilon n^{1-O(1/c^2)}$ (suppressing some low order terms). Note that the decay rate of the additive errors is worse than linear in $n$. In comparison, for Gaussian KDE (even with shuffled DP) we get a true additive approximation, and with linear error decay rate $1/\varepsilon n$.
>
> Second, in terms of efficiency, their mechanism seems infeasible for shuffled DP. It maintains $n^{\ln\ln n}$ counters (corresponding to nodes of a $T$-ary tree of depth $K$, where $K,T\sim\ln n$). In their central DP setting, to achieve efficiency, they round all small counters to zero, and realize only the non-zero counters, of which there are at most $\widetilde O(n)$. But with shuffled DP, each user would have to participate in a summation protocol for each of the $n^{\ln\ln n}$ counters (the users cannot know ahead of time which counters would end up small), leading to infeasible (worse than any polynomial) running time and communication per user.
>
> [A] Gursoy, M.E., Inan, A., Nergiz, M.E. and Saygin, Y., Differentially private nearest neighbor classification. Data Mining and Knowledge Discovery, 2017.
>
> [B] Zhu, Y., Yu, X., Chandraker, M. and Wang, Y.X., Private-knn: Practical differential privacy for computer vision. CVPR 2020.
>
> [C] Andoni, A., Indyk, P., Mahabadi, S. and Narayanan, S., Differentially private approximate near neighbor counting in high dimensions. NeurIPS 2023

---

### Official Review · Reviewer_133G · 2024-11-05

**Soundness:** 3
**Presentation:** 3
**Contribution:** 3
**Rating:** 8
**Confidence:** 3

**Summary:**

This paper proposes a new method to perform kernel density estimation (KDE) under Shuffle model of differential privacy (Shuffle-DP).

The KDE problem is as follows: Given datapoints $X = (x_1, \ldots, x_n)$ and a query point $y$ (all points in $\mathbb{R}^d$), the goal is to compute $\mathrm{KDE}\_X(y) := \frac1n \sum\_{i=1}^n \mathbf{k}(x\_i, y)$ for a given kernel $\mathbf{k}$.

The Shuffle-DP is the setting where messages from the users are sent to a shuffler that randomly permutes all received messages before sending it to the analyst, and it is required that the multi-set of messages received by the analyst satisfies differential privacy.
Various Shuffle-DP mechanisms are known in literature for computing "Bit-sums", wherein, each user holds a single bit $x_i \in \{0, 1\}$ and the goal is to estimate their sum $\sum_i x_i$.

The main technique in this paper is to reduce the problem of KDE to Bit-sums, for kernels $\mathbf{k}$ that admit "(approximate) locality sensitive quantization (LSQ)". This means that there is a distribution $\mathcal{Q}$ over pairs of functions $(f, g)$ each mapping $\mathbb{R}^d \to [-R, R]^Q$ such that $\mathbb{E}_{(f, g) \sim \mathcal{Q}} [f(x)^T g(y)]$ is equal to (or closely approximates) $\mathbf{k}(x, y)$ for all $x, y \in \mathbb{R}^d$.

Thus, given access to $f_i(x_j)$ for $i \in \{1, \ldots, I\}$ and all $x_j \in X$, one can approximate $\mathbf{k}(x, y)$ as $\frac1I \sum\_{i=1}^I \sum\_{x \in X} f\_i(x)^T g\_i(y)$.

Finally, the $f_i(x_j)$ themselves are estimated by reducing to "Bit-sums" through randomized rounding.

Finally it is standard to use KDE for solving classification tasks, by performing a KDE on examples of each class, and on given query point $y$, return the class with the largest kernel density estimate.

The paper provides bounds on the communication cost and root-mean-squared-error (worst case over all query points $y$).

The paper also provides experimental evaluation on three textual datasets (two on topic classification, one on sentiment classification) and an image classification dataset (CIFAR-10), and evaluates the performance of two kernels (Gaussian kernel and Inner Product kernel) in three regimes: (i) without privacy, (ii) central DP and (iii) shuffled DP.

**Strengths:**

The paper proposes a novel application of Bit-sums in Shuffle-DP to the problem of Kernel Density Estimation. The idea is simple and easy to implement. The paper has rigorous theoretical bounds on the error.

Overall the paper is well written and easy to read.

**Weaknesses:**

This is not particularly a "weakness", but the approach in the paper is limited to kernels that admit good locality sensitive quantization.

**Questions:**

I would imagine that the theoretical bounds on the $\mathrm{supRMSE}$ in Theorem 3.2 are quite loose. Would it be possible to add a comparison of the theoretical bounds alongside the empirically realized errors in KDE ?

For example, I am surprised that for the IP kernel, it is better to go with the $(1, \sqrt{d}, 1)$-LSQ instead of the naive $(d, 1, d)$-LSQ. While the latter has additional dimension, it is no variance, whereas the former has a large variance, even if it is $1$ dimensional. But I don't see the bounds in Theorem 3.2 accounting for the variance in the LSQ?

---

> ### Author Response · Authors · 2024-11-18
>
> We thank you for the review.
>
> > I would imagine that the theoretical bounds on the supRMSE in Theorem 3.2 are quite loose. Would it be possible to add a comparison of the theoretical bounds alongside the empirically realized errors in KDE ?
>
> We have added a comparison in the uploaded version (Figure 5, page 24). The looseness of the theoretical error bound depends on the data: for some datasets it is quite tight, while for others it is rather pessimistic.
>
> > I don't see the bounds in Theorem 3.2 accounting for the variance in the LSQ?
>
> The variance comes into play in the dependence of the supRMSE on $R,S$ in Theorem 3.2, which is $SR^2/\sqrt{I}$ over $I$ repetitions. The reason is largely that this is the standard deviation of averaging $I$ i.i.d repetitions of a random variable bounded in an interval of length $SR^2$. The random variable here is $\tfrac1n\sum_{i=1}^nf(x))^Tg(y)$ for a sample $(f,g)\sim\mathcal Q$, and it is bounded in $[-SR^2,SR^2]$ because the inner product has $S$ non-zero summands due to sparsity, each a product of two numbers in $[-R,R]$ (the corresponding entries of $f$ and of $g$). In the proof, this is happening in Section A.1.4, particularly Claim A.4 and the bound after it.
>
> > I am surprised that for the IP kernel, it is better to go with the $(1,\sqrt d,1)$-LSQ instead of the naive $(d,1,d)$-LSQ. While the latter has additional dimension, it is no variance
>
> In the shuffled DP KDE protocol, apart from the randomness of sampling from the LSQ family, there is another source of randomness, due to discretization (which is necessary for bounding the bit-width). Since the random discretization error also yields the same $SR^2/\sqrt{I}$ dependence on $R,S$, the mechanism incurs this error even if the LSQ family is random-free. Technically, in the proof, this would make the expectations in Claim A.2 and in Section A.1.4 be over only $\{b_{ij}^{(u)}\}$ (the randomness of discretization), and not also over $(f_i,g_i)$ (the randomness of LSQ) as they are currently, but otherwise the arguments and the bounds they yield remain the same. Without discretization (for example in central DP), you are correct that the random-free family would lead to lower KDE error. However, the resulting mechanism would not be implementable with shuffled DP.
>
> Under shuffled DP, since with both the $(d,1,d)$-LSQ and $(1,\sqrt{d},1)$-LSQ for IP we have $SR^2=d$, the error in both cases is similar. The advantage of the $(1,\sqrt{d},1)$-LSQ is its much smaller $Q$, which doesn’t affect the error to privacy trade-off, but allows better control of the running times and the communication cost.
>
> We have added to the uploaded version a comparison of the KDE error of these two LSQs for the IP kernel (Figure 6, page 24). The results indicate that their empirical error is very similar.

---

> > ### Comment · Reviewer_133G · 2024-11-22
> > **Acknowledgement**
> >
> > I thanks the authors for the response. I will keep my rating.

---

### Official Review · Reviewer_A2xf · 2024-11-08

**Soundness:** 3
**Presentation:** 2
**Contribution:** 3
**Rating:** 6
**Confidence:** 3

**Summary:**

This paper studies kernel density estimation (KDE) under shuffle DP, which is suitable for the setting where users do not persistently participate in training. The authors prove convergence for Gaussian kernel and test their algorithm on various datasets.

**Strengths:**

1. Kernel density estimation is an important machine learning problem, and shuffle DP is an important intermediate privacy model between local and central DP.
2. The authors conducted extensive experiments on various datasets.

**Weaknesses:**

1. The algorithm proposed in this work mainly seems like an application of bitsum algorithm. The authors should highlight the novelty in either the algorithm design or theoretical analysis. For example, what is the main difficulty in applying to kernel density estimation
2. The algorithm does not demonstrate a consistent good performance. The performance is bad for inner product kernel and shuffle-DP algorithms other than 3NB (Bhadi et.al. 2020).
3. The authors claimed that there is no need to optimize the bandwidth for the Gaussian kernel. However, by shrinking the dataset by some factor, one can equivalently view this as changing the bandwidth. Thus, this claim may not be justified.

======= Update =======
Increased my score to 6 after the author's response.

**Questions:**

How can the algorithm adapt to new users joining in and we need to update the kernel density?

---

> ### Author Response · Authors · 2024-11-18
>
> We thank you for the review.
>
> > what is the main difficulty in applying to kernel density estimation
>
> Let us highlight the difficulties. Generally, it is not the case that every central DP mechanism can be turned into shuffled DP by replacing its summation operations with shuffled DP sum protocols. Rather, this depends on the specific mechanism. The two main barriers are,
> * While central DP mechanisms use explicit noise distributions (like Gaussian or Laplace noise), these distributions are generally not realizable with shuffled DP summation protocols. These protocols yield different noise distributions, which are mostly given non-explicitly, but only as a single statistic (like the RMSE). The mechanism needs to be re-designed and re-proven under those more difficult distributions, and this generally may be difficult or impossible, depending on how the central mechanism utilizes specific properties of its original explicit distributions.
> * Shuffled DP sums force errors to be introduced at every summation operation, unlike the central mechanism, which can compute exact sums and introduce noise at other more convenient stages in the mechanism. This creates a new source of error, which accumulates in a way that depends on how the specific central mechanism combines the sums, and needs to be controlled and bounded in order to prove a utility guarantee under shuffled DP.
>
> This is why we have to “open the black-box”, and re-analyze the DP KDE mechanism from first principles in Theorem 3.2, rather than using the existing results from central DP. The proof yields a utility guarantee from much weaker conditions than the central case.
>
> Concretely, in the central DP KDE setting (Wagner et al., 2023), once the notion of LSQ is introduced, the mechanism itself amounts to the basic Laplace mechanism over the LSQ features. This mechanism has known subexponential noise behavior and a built-in utility guarantee in terms of the $\ell_1$-sensitivity. In the shuffled DP case, no such generic off-the-shelf mechanism is available, and the analysis of Theorem 3.2 proves an alternative result specifically for KDE, by analyzing moments of the unspecified distribution of noise that accumulates from bitsums and discretization through the LSQ computation.
>
> Discretization (from real shuffled DP sums to bitsums) is a third source of difficulty, which is necessary to control the bit-width of the mechanism (a notion that does not exist in central DP). In the central KDE mechanism, the LSQ features $f(x)$ (for a given pair $(f,g)\sim\mathcal Q$) are a function of the data point $x$. Under our randomized discretization, they are the result of a randomized computation (thus, two users with the same input and same $f$ would still produce different features). This randomness introduces another source of error that does not exist in the central setting, and needs to be controlled throughout the proof of Theorem 3.2.
>
> As a “negative example”, please see the discussion of DP NNC in the answer to 3tVM. This is a problem related to DP KDE, whose existing central DP mechanism seems non-adaptable to shuffled DP, for (some of) the above reasons, even though all its summation operations are bitsums.
>
> > The authors claimed that there is no need to optimize the bandwidth for the Gaussian kernel. However, by shrinking the dataset by some factor, one can equivalently view this as changing the bandwidth. Thus, this claim may not be justified.
>
> This setting of bandwidth is based on the vectors being normalized to unit length (this was stated in line 346, and in the uploaded version we also clarify this in the context of the bandwidth setting as well). You are correct that if the data is scaled, the bandwidth should be scaled accordingly.

---

> > ### Author Response · Authors · 2024-11-18
> >
> > > How can the algorithm adapt to new users joining in and we need to update the kernel density?
> >
> > The KDE estimation part of the algorithm naturally supports new users, due to the linearity of KDE (we can just add the estimates with proper scaling). The issue is that the shuffled DP model itself isn’t suited for new users: once the shuffler was invoked on the initial set of users, meaning it produced a random permutation of their messages and forwarded it to the analyzer, new messages from new users cannot be retroactively mixed into the permutation (the analyzer would necessarily know that the new users came last).
> >
> > New users can still be added, by protecting themselves with local DP (which is the special case of shuffled DP with a single user), at the cost of degrading the overall accuracy of the combined KDE estimate. Concretely, for the Gaussian kernel, suppose we have $n$ initial users who participate in the shuffled DP protocol. Their DP KDE supRMSE in Theorem 3.3 decays like $1/n$. Now suppose $m$ new users arrive one by one, each protecting herself with local DP (with the same $\varepsilon$ privacy budget). The combined KDE function remains $\varepsilon$-DP, and its supRMSE now decays like $\sqrt{m}/(n+m)$. If few users are added (say, $m=O(1)$) the decay is still linear in the total number of users. As the number of new users grows and overwhelms the original number ($m\gg n$), the error decay degrades to square root of the total number of users. This square-root error decay rate is the same we would get by just using local DP to begin with, without a shuffler.
> >
> > On the other hand, if the $m$ new users come as a batch, they can run their own separate shuffled DP protocol, and combine its KDE with the original KDE. In that case, the total supRMSE does decay linearly, like $1/(m+n)$, even for large $m$.
> >
> > Both of these are special cases of the following: suppose the users arrive in $B$ sequential batches of sizes $n_1,\ldots,n_B$, and each batch run their own shuffled DP protocol (which for batches of size $n_i=1$ means just using local DP), and combine its KDE with the previous ones. Then, the total KDE supRMSE decay rate is $\sqrt{B}/\sum_{i=1}^B n_i$. Thus, the more the users “batch” together (in less batches) the better, since that way they can better exploit the accuracy boost of shuffled DP compared to local DP.

---

> > > ### Comment · Reviewer_A2xf · 2024-11-26
> > >
> > > Thanks for your thoughtful response. I will increase my score to 6 if no other major issues are found.

---

### Author Response · Authors · 2024-11-18
**We have posted initial responses to the reviews**

We thank all the reviewers for their engaging and detailed reviews and for the valuable comments and suggestions. We have posted responses to each review. We have uploaded an updated PDF containing some additional experiments as requested.

(Edit: New PDF uploaded with corrections to typos and formatting)

---

### Meta-Review · Area_Chair_gKbf · 2024-12-14

**Metareview:**

## Summary of Contributions

The paper considers the kernel density estimation (KDE) problem. Here, for a dataset $x_1, \dots, x_n$, a kernel $k$ and a query $y$, the goal is to estimate $\frac{1}{n} \sum_{i \in [n]} k(x_i, y)$. We want to do this while protecting privacy of the dataset. The paper works in the *shuffle differential privacy (SDP)* model, where each user $i$ holds $x_i$ and can produce arbitrary messages. The messages are then randomly permuted, and the DP guarantee is enforced on these shuffled messages. This is a distributed model of DP that interpolates between central-DP and local-DP. The authors give an SDP algorithm for KDE. The rough idea is to use a notion of locality sensitive quantization (LSQ), which allows one to compute $k(x, y)$ based on $E[\left<f(x), g(y)\right>]$ where $f, g$ are (random) vector-valued functions. This essentially allows us to "linearize" the computation: The goal is now essentially to estimate $\sum_{i \in [n]} f(x_i)$. This is just a vector summation problem, and the authors give an efficient protocol for this via known SDP bit-sum protocols in literature. They then prove that the algorithm achieves small asymptotic errors (nearly matching central-DP for certain kernels) and show that the algorithm performs well empirically.

## Strengths

- KDE is a fundamental problem and this paper improves our understanding of private KDE in the shuffled model.

- The protocol proposed is simple, elegant and can be practical. Furthermore, the errors are shown to be asymptotically similar to those of the central DP protocols.

- The experiments show promising results; the algorithms proposed here can nearly match the accuracy of central DP algorithms for moderate values of $\epsilon$ on most datasets.

## Weaknesses

- The approach is limited to kernels that admit good LSQs.

- Despite the asymptotic optimality, the experiments still show a significant gap between the algorithm here and the central DP one in small $\epsilon$ regime (which is arguably more important) and for some datasets (e.g. CIFAR-10).

**Additional Comment**: The current title is a little bit too vague. Given that the paper focuses on KDE, we suggest that this is mentioned somehow in the title.

**Additional Comments On Reviewer Discussion:**

During the discussion, the authors have clarified with regards to the difficulty in applying shuffle-DP to other non-private algorithms, and the distinction between classification and class decoding. They also clarify comparisons with other models (local-DP and central-DP) both in terms of theoretical guarantees and experiments.

---

> ### Public Comment · ~Tal_Wagner1 · 2025-02-18
>
> Paper title revised, as per the meta-review recommendation. Thank you for the feedback.

---

### Decision · Program_Chairs · 2025-01-22

Accept (Spotlight)